*A Nature Portfolio journal*

# ATM Inhibition Enhances Knock-in Efficiency by Suppressing AAV-Induced Activation of Apoptotic Pathways
Munkh-Erdene Natsagdorj [1,8], Hiromasa Hara [1,2,3,8] ✉, Hideki Uosaki [1,4,5], Fumio Nakahara[1], Makoto Inoue [2,6,7] & Yutaka Hanazono [1,2,4] ✉

CRISPR/Cas9-mediated genome editing has expanded the possibilities for precise gene modifications; however, the efficiency of targeted insertion remains suboptimal. In this study, we describe a triple-reporter system in mouse embryonic stem cells that simultaneously tracks double-strand break (DSB) induction, homology-directed repair (knock-in), and end-joining-mediated targeted insertion (EJ-TI). Using both plasmid and adeno-associated virus (AAV) donor vectors, our results demonstrate that ataxia telangiectasia and Rad3-related kinase (ATR) activity is essential for knock-in regardless of the donor type, whereas ataxia telangiectasia mutated (ATM) inhibition exhibits a donor-dependent role. In cells receiving circular plasmid donors, ATM inhibition with AZD1390 markedly reduced the knock-in and EJ-TI efficiencies, consistent with its canonical role in DSB repair. In contrast, with linear AAV donors, ATM inhibition enhanced the knock-in efficiency by suppressing the overactivation of the ATM-p53-caspase 3 apoptotic pathway and partially suppressing classical non-homologous end-joining. These findings highlight the critical influence of donor DNA configuration on DNA damage response signaling and provide a strategy for optimizing genome editing efficiency by selectively modulating the ATM pathways, an approach that may have significant implications for gene therapy, cell engineering, and other applications.

Genome-editing technologies such as CRISPR/Cas9 enable the induction of double-strand breaks (DSBs) at specific genomic loci[1–4]. These technologies have greatly improved the efficiency of targeted insertion of exogenous DNA[5] into the genome via homology-directed repair, a process known as "knock-in", and have unlocked a wide range of applications in medicine, agriculture, and aquaculture[6]. One of the major challenges in genome editing is the enhancement of knock-in efficiency. As an error-free pathway, knock-in is particularly crucial in medical applications where high precision is of utmost importance. Although numerous strategies have been reported to enhance knock-in efficiency[7–9], they have been hindered by limited effectiveness and inadequate reproducibility[10]. Because experimental conditions vary widely among studies, developing a universally effective approach remains challenging.

In recent years, adeno-associated virus (AAV) has emerged as promising donor vectors for knock-in because of their ability to mediate knock-in with relatively high efficiency and minimal cytotoxicity[11–14]. Unlike plasmid donor vectors, which exist as double-stranded DNA (dsDNA), AAV contains single-stranded DNA (ssDNA)[15]. Because distinct sensors within the DNA damage response (DDR) network detect DSBs and single-strand breaks (SSBs)[16], they can initiate specific DDR pathways based on the structural properties of the donor DNA. This potential variation in DDR activation suggests that AAV may elicit a unique, AAV-specific DDR. A comprehensive understanding of the molecular mechanisms governing

[1]Division of Regenerative Medicine, Center for Molecular Medicine, Jichi Medical University, 3311-1 Yakushiji, Shimotsuke, Tochigi, 329-0498, Japan. [2]Laboratory of Regenerative and Cellular Medicine, Center for Development of Advanced Medical Technology, Jichi Medical University, 3311-1 Yakushiji, Shimotsuke, Tochigi, 329-0498, Japan. [3]Division of Animal Resource Development, Center for Development of Advanced Medical Technology, Jichi Medical University, 3311-1 Yakushiji, Shimotsuke, Tochigi, 329-0498, Japan. [4]Division of Translational Research, Center for Development of Advanced Medical Technology, Jichi Medical University, 3311-1 Yakushiji, Shimotsuke, Tochigi, 329-0498, Japan. [5]Division of Functional Biochemistry, Department of Biochemistry, School of Medicine, Jichi Medical University, 3311-1 Yakushiji, Shimotsuke, Tochigi, 329-0498, Japan. [6]Sumitomo Pharma Co., Ltd., 2-6-8 Doshomachi, Chuo-ku, Osaka, 541-0045, Japan. [7]Racthera Co., Ltd., 2-7-1 Nihonbashi, Chuo-ku, Tokyo, 103-6012, Japan. [8]These authors contributed equally: Munkh-Erdene Natsagdorj, Hiromasa Hara. ✉e-mail: hhara@jichi.ac.jp; hanazono@jichi.ac.jp

AAV-mediated knock-in is essential for refining its application in genome editing to improve its precision, efficiency, and therapeutic potential.

Despite significant advances in genome editing, current methods for evaluating editing efficiency remain limited in terms of accuracy and resolution. Conventional approaches, such as PCR-based assays[17–19], which are subject to amplification biases, and fluorescent reporter systems[20,21], which exhibit limited ability to reliably and quantitatively distinguish precise knock-in events from error-prone end-joining-mediated targeted insertions (EJ-TI), have inherent methodological limitations. These limitations hinder not only the accurate quantification of genome-editing outcomes but also the comprehensive analysis of the underlying repair mechanisms.

In this study, we developed a triple-reporter system that simultaneously monitors DSB induction, knock-in, and EJ-TI. Using both plasmid and AAV donor vectors, we conducted screening using a comprehensive DDR inhibitor library to identify the key factors influencing knock-in efficiency. The screening revealed that ataxia telangiectasia mutated (ATM) inhibition exhibited a donor-dependent effect, diminishing the knock-in efficiency with plasmid donors while enhancing it with AAV donors. These findings provide valuable insights into how donor DNA structure influences DDR signalling and establish a foundation for improving knock-in efficiency.

## Results

### Establishment of triple-reporter embryonic stem cells

To overcome the limitations of conventional assays that assess each genome editing outcome separately, we established a triple-reporter system to simultaneously monitor DSB induction, knock-in, and EJ-TI events. First, we established a DSB reporter using TagBFP. A gRNA was designed to cleave the tyrosine within the TagBFP chromophore (Fig. 1a). TagBFP was heterozygously inserted to fuse with the C-terminus of *Actb* in mouse embryonic stem (ES) cells (Fig. 1b, Supplementary Fig. 1a). As shown in Supplementary Fig. 1b, TagBFP fluorescence was diminished after the electroporation of a plasmid expressing Cas9 and gRNA. The surveyor assay confirmed that the diminished TagBFP fluorescence reflected mutations in the TagBFP chromophore induced by Cas9 and the gRNA (Supplementary Fig. 1c). Because tyrosine is encoded only by TAC and TAT, any mutation other than cytosine-to-thymidine substitution results in the loss of TagBFP fluorescence. Deep sequencing further demonstrated that TagBFP efficiently detected both frame-shift and in-frame mutations (Fig. 1c, Supplementary Fig. 1d).

Next, we established a targeted insertion reporter (TI reporter) to simultaneously detect both knock-in and EJ-TI events using mCherry and mEGFP in mouse ES cells. In this system, the reporter cells contained sequences identical to those recognized by the gRNA for cleaving TagBFP in the DSB reporter, in addition to a splice acceptor and mCherry (Fig. 1d). These cells use the splice acceptor to translate lacZ–mCherry downstream of *Actb* exons 1–3. After transfecting TI reporter cells with plasmids expressing Cas9 and gRNA along with a donor vector, mEGFP was inserted upstream of mCherry. Under knock-in conditions, both mCherry and mEGFP were expressed (Fig. 1e, knock-in). Under EJ-TI conditions, two splice acceptors are available: a genomic splice acceptor embedded in the reporter locus upstream of a stop codon, and a donor-derived splice acceptor positioned immediately upstream of mEGFP within the donor cassette (Fig. 1e, EJ-TI). Splicing to the genomic splice acceptor yields a non-productive isoform (no mEGFP) because a stop codon is located upstream of mEGFP. In contrast, when splicing occurs at the donor-derived splice acceptor, the upstream stop codon is skipped in the mature transcript, resulting in mEGFP expression. Consistent with mixed splice acceptor usage within the same cells, mEGFP fluorescence in EJ-TI was lower than in knock-in (Fig. 1h). On the other hand, because the donor cassette contains a stop codon at its 3′ end, translation of mCherry is terminated, resulting in loss of its expression in EJ-TI. We established cells carrying the TI reporter in one allele of *Actb*, whereas the other allele remained intact (Supplementary Fig. 2a). To test the TI reporter system, we examined donor plasmid

vectors with 1-kb arms, with or without HITI sequences (pReporter–donor–HITI or pReporter–donor, respectively) at both ends for linearization[22]. Donor plasmids and a plasmid expressing Cas9 and gRNA were electroporated into the TI reporter cells. Flow cytometry and fluorescence microscopy identified four distinct fractions: mCherry single-positive, mEGFP single-positive, double-positive, and double-negative cells (Supplementary Fig. 2b, c). Each fraction was then sorted and subjected to PCR. Only the double-positive fraction yielded knock-in-specific bands, whereas wild-type bands were observed in the mCherry single-positive and double-negative fractions (Fig. 1f). This finding suggests that the wild-type bands observed in the mCherry single-positive and double-negative fractions resulted from either no editing or insertion/deletion (InDel) mutations at the targeting site. For the mEGFP single-positive fraction, the PCR band sizes using pReporter–donor–HITI were similar to those expected for HITI-excised constructs inserted via end-joining. In contrast, when pReporter–donor was used, a much larger band was detected in the mEGFP single-positive fraction compared with pReporter–donor–HITI. The larger band corresponds to full-length plasmid DNA, including the backbone, which is likely integrated via end-joining after linearization by cellular endonucleases. This suggests that the pReporter–donor, which is linearized by cellular endonucleases, is integrated into the locus via end-joining. Taken together, the results showed that our TI reporter system can detect both knock-in and EJ-TI by flow cytometry. This reporter offers significant advantages in terms of accuracy, sensitivity, and simplicity compared with the conventional PCR-based method, in which shorter PCR products are dominantly amplified (Supplementary Fig. 3).

To investigate whether mCherry diminishing could detect mutations caused by DSBs, cells were transfected with plasmid DNA expressing Cas9 and gRNA. Deep sequencing revealed that nearly all cells in the mCherry-negative fraction harbored frame-shift mutations, whereas in-frame mutations were predominantly found in the mCherry-positive fraction (Supplementary Fig. 4). Therefore, the TagBFP system, which detects both frame-shift and in-frame mutations, is superior to the mCherry system for DSB detection.

To simultaneously monitor DSB, knock-in, and EJ-TI events, we established triple-reporter ES cells by combining the DSB reporter (Actb–TagBFP) with the TI reporter (Actb–lacZ–mCherry) (Fig. 1g, Supplementary Fig. 5a, b). The detection power of these triple-reporter cells was comparable to that of cells in which only one system was introduced (Fig. 1h, Supplementary Fig. 5c). In TagBFP-positive cells, no targeted insertion was observed, and mCherry diminishing was rare (Fig. 1h). In contrast, TagBFP-negative cells exhibited targeted insertion, confirming that the DSB reporter accurately reflects DSB induction at the TI reporter.

Further analysis using the pReporter–donor revealed the specific repair pathways contributing to each fraction (Supplementary Fig. 6). DSB repair, indicated by the TagBFP-negative fraction, was selectively impaired by DNA-PK inhibition, whereas RAD51 and PARP1 inhibition had no effect. These results indicate that classical non-homologous end-joining (c-NHEJ) predominates under these conditions, with minimal contributions from homologous recombination (HR) and alternative end-joining (alt-EJ). In contrast, the knock-in efficiency decreased only upon RAD51 inhibition, while neither DNA-PK nor PARP1 inhibition had an effect, confirming their exclusive dependence on RAD51-mediated HR. Conversely, the decrease in EJ-TI events observed with either RAD51 or PARP1 inhibition (Supplementary Fig. 6c, d) suggested that both RAD51-mediated HR and PARP1-dependent alt-EJ, although mechanistically distinct, may collaboratively contribute to these events.

### DDR-inhibitor screening distinguishes donor-dependent modulators of targeted insertion

To identify key DDR pathways that modulate knock-in depending on donor vector type, we conducted a drug screening using the DNA Damage/DNA Repair Compound Library (Supplementary Data 1; 174 compounds) under two conditions: (i) plasmid donor (pReporter-donor) with a plasmid

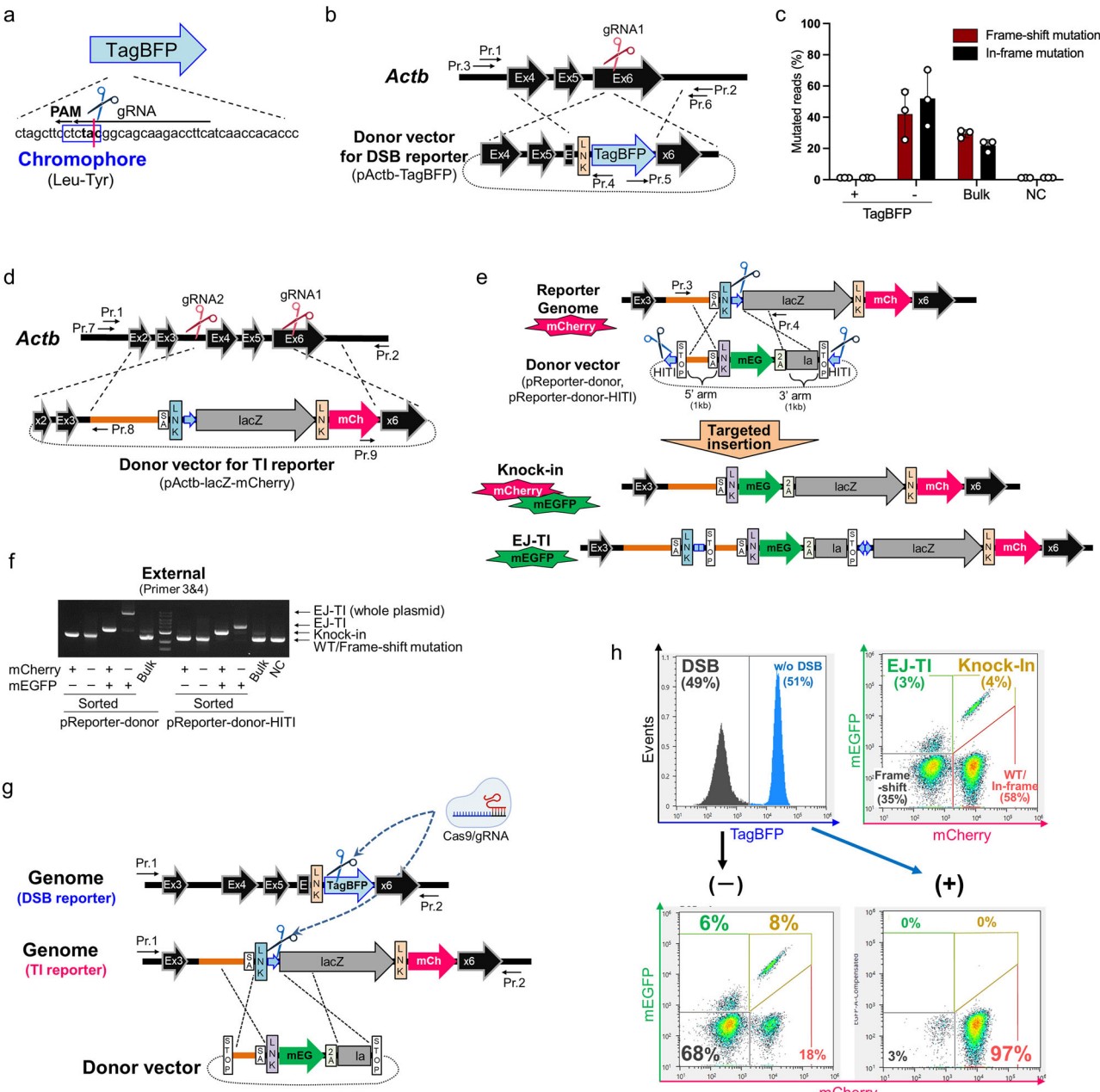

**Fig. 1 | A triple-reporter system for the simultaneous detection of DSB induction, knock-in, and end-joining-mediated targeted insertion (EJ-TI). a–c** Detection of frameshift and in-frame mutations using the DSB reporter. **a** Schematic representation of the DSB reporter system, in which a gRNA targets the tyrosine residue within the TagBFP chromophore, potentially disrupting fluorescence. **b** Diagram of the reporter construct, where TagBFP is fused to the C-terminus of *Actb* in mouse embryonic stem (ES) cells. "LNK" denotes a linker sequence; the same notation is used in panels "**d**", "**e**", and "**g**". **c** Deep sequencing analysis of genome-edited populations after plasmid transfection expressing Cas9 and gRNA. "NC" represents non-genome-edited control cells, while "Bulk" represents genome-edited but unsorted cells. The proportion of frame-shift (red) and in-frame (black) mutations is shown. Data are presented as mean ± SD. (*n* = 3 independent experiments). **d–f** The targeted insertion reporter differentiates between knock-in and EJ-TI. **d** Schematic representation of the targeted insertion (TI) reporter system. A lacZ–mCherry cassette was inserted into *Actb* using CRISPR/Cas9, driven by a splice acceptor to enable expression alongside *Actb*. **e** Diagram illustrating the distinction between

knock-in and EJ-TI events. The donor vector introduces mEGFP upstream of mCherry. Knock-in results in the expression of both mEGFP and mCherry, whereas EJ-TI leads to mEGFP expression without mCherry. **f** PCR analysis of the fluorescence-sorted cell populations. Wild-type (WT) bands were detected in the mCherry single-positive and double-negative fractions. Knock-in-specific bands were observed in the double-positive fraction, whereas EJ-TI bands were detected in the mEGFP single-positive fraction. **g, h** Establishment of a triple-reporter system for the simultaneous detection of DSB and targeted insertion events. **g** Schematic representation of the system integrating DSB and TI reporters. **h** Flow cytometric analysis of genome editing outcomes in triple-reporter ES cells after transfection of plasmid vectors both expressing Cas9 and gRNA, and pReporter–Donor. DSB events are identified by the loss of TagBFP fluorescence (TagBFP-negative fraction). The restriction of knock-in and EJ-TI events to the TagBFP-negative population confirms the coordination between DSB and TI reporters. The illustrations of scissors and Cas9 were created with BioRender.com.

expressing Cas9/gRNA and (ii) AAV donor with Cas9/ribonucleoprotein (RNP). Screening outcomes were quantified with an established triple-reporter system in mouse ES cells, and hits were annotated relative to the mean value of the untreated control, using the average value plus or minus three standard deviations (Average ± 3 SD) as the thresholds (Supplementary Fig. 7).

We first noted that individual PARP inhibitors exhibited compound-specific and inconsistent activity: rucaparib phosphate (#49) showed an increase above the Average + 3 SD in the plasmid-Cas9/plasmid-donor condition, whereas other PARP inhibitors (e.g., niraparib #33; AZD2461 #157) did not reproduce this effect with either plasmid or AAV donors, suggesting that the response was likely compound-specific or off-target rather than a robust enhancer. Therefore, we deprioritized PARP for further validation. ATR inhibitors (VE-821 #144; BAY 1895344 #151) consistently decreased knock-in efficiency under both plasmid- and AAV-donor conditions. In contrast, ATM inhibitors (KU-55933 #37; KU-60019 #53) exhibited the opposite donor-type dependency, decreasing knock-in with plasmid donors but causing little or modest increases with AAV donors. Taken together, although ATR and ATM are two closely related members of the phosphatidylinositol 3-kinase–related kinase family central to DDR, our screen revealed divergent donor-type behavior: ATR acted irrespective of donor type, whereas ATM showed donor-type dependence. Accordingly, we focused subsequent validation analyses on ATR and ATM.

## Opposing effects of ATM inhibitors on targeted insertion depending on donor vector type, but not observed with ATR inhibition

Guided by these screening results that identified ATR and ATM as key candidate regulators, we next validated their donor-type-dependent effects on targeted insertion using the triple-reporter system under plasmid- and AAV-donor conditions. To enhance our analysis, we selected the most potent inhibitors from a broader inhibitor panel and optimized their concentrations for subsequent experiments.

We first investigated ATR. ATR is recruited to RPA-coated ssDNA and activated during the DDR[23]. The AAV genome consists of ssDNA and has been shown to induce ATR phosphorylation upon transduction[24]. Thus, ATR inhibition might differentially influence cellular responses depending on donor vector type, as AAV donors contain ssDNA whereas plasmid donors contain dsDNA. However, contrary to our expectations, ATR inhibition with either VE-821 or ceralasertib significantly decreased the knock-in efficiency for both plasmid and AAV donors (Fig. 2a-c). Because ATR inhibition impaired knock-in regardless of vector type, we hypothesized that stimulating ATR might increase the knock-in efficiencies. NRF2 promotes ATR phosphorylation[25] but is kept at low levels by KEAP1-mediated ubiquitinuation, so we used 4-octyl itaconate (4OI) to inhibit KEAP1 and stabilize NRF2 (Fig. 2d). Treatment with 4OI increased NRF2 protein levels and enhanced phosphorylation of both ATR and its downstream effector CHK1 following DNA damage (Fig. 2e). Consistent with our rationale, 4OI significantly increased knock-in efficiencies for plasmid donors (11.1 ± 0.2% vs. 9.7 ± 0.4%) and AAV donors (46.9 ± 2.9% vs. 20.4 ± 1.3%) (Fig. 2f, g). Together, these results indicate that ATR activity promotes targeted insertion regardless of donor type and acts as a positive regulator of knock-in efficiency.

We then evaluated ATM. When using plasmid donor vectors, treatment with ATM inhibitors, whether AZD1390 or AZ31, significantly decreased the efficiencies of both knock-in and EJ-TI (Fig. 3a). These results indicate that ATM plays a critical role in facilitating targeted insertion with plasmid donor vectors. In contrast, when AAV donor vectors were used, treatment with ATM inhibitors increased the efficiencies of both knock-in (24.4%–25.3% vs. 18.3%) and EJ-TI (3.4%–3.9% vs. 2.1%, Fig. 3b). To evaluate whether this effect extended beyond mouse ES cells, we applied the same ATM inhibitor treatment to human embryonic kidney-derived cells (HEK293T) and immortalized mesenchymal stromal cells (MSC). In both cell lines tested, ATM inhibition reproducibly increased knock-in

efficiencies (Fig. 3c), demonstrating that ATM suppression broadly potentiates AAV-mediated knock-in across diverse mammalian cells.

## Linearized vs. circular donor structures drive different ATM inhibitor responses

To investigate whether the differential response to ATM inhibition between the plasmid and AAV donor vectors was due to differences in DNA structure (ssDNA vs. dsDNA), we first tested self-complementary AAV (scAAV), a double-stranded form of AAV. scAAV exhibited the same response as AAV (Fig. 4a), suggesting that strand type was not a determining factor. Further supporting this, linearized ssDNA also showed the same response as AAV (Fig. 4b), reinforcing the conclusion that the difference in response to ATM inhibition was independent of whether the donor DNA was single-stranded or double-stranded. Next, we considered whether the structural form (linear vs. circular) could influence the response. To test this, we used a donor vector that was excised from a circular plasmid (pReporter–donor) via restriction enzyme digestion to generate a linearized dsDNA donor. This linearized dsDNA donor exhibited the same response to ATM inhibition as AAV (Fig. 4c). In addition, the circular plasmid donor flanked by HITI sequences (pReporter–donor–HITI), which was cleaved by Cas9 to generate linearized dsDNA at both homology arm ends, also showed the same response to ATM inhibition as AAV (Fig. 4d). These findings indicate that the structural form of the donor—linear versus circular—determines the differential response to ATM inhibition.

## Reduction of knock-in efficiency by AAV-induced activation of apoptotic and c-NHEJ pathways

To investigate why ATM inhibitors have different effects depending on the donor vector type, we first compared the impact of plasmid and AAV donor vectors on ATM activation. When an AAV donor or a plasmid donor with HITI (pReporter–donor–HITI) was used, phosphorylated ATM levels increased markedly, whereas a circular plasmid donor (pReporter–donor) induced only a slight increase (Fig. 5a). These results indicate that the transfection of linear DNA triggers ATM activation. To further isolate the effect of the donor vector itself, independent of potential interference from Cas9/RNP, we conducted experiments using only the donor vector, excluding Cas9/RNP delivery (Fig. 5b, c). Western blotting confirmed that total ATM levels did not vary under any condition. In contrast, phosphorylated ATM levels were markedly elevated following AAV transduction, whereas transfection with a plasmid donor did not induce significant changes. In addition, transduction with empty AAV capsids failed to trigger ATM activation, indicating that the presence of AAV-derived DNA is necessary for ATM activation. Downstream components of the ATM signaling pathway, including p53 and caspase 3, were activated exclusively following AAV transduction. ATM inhibition with AZD1390 reduced the levels of phosphorylated ATM, phosphorylated p53, and active caspase 3 to baseline values comparable to those observed in the negative control (Fig. 5d, e). Indeed, treatment with the ATM inhibitor significantly reduced the proportion of early apoptotic cells (Annexin V⁺ among Zombie NIR⁻ cells) after genome editing (Fig. 5f). To further elucidate the role of the p53–caspase 3 pathway in mediating the effect of ATM inhibition on knock-in, we assessed the impact of a p53 inhibitor. As expected, treatment with pifithrin-μ resulted in a significant increase in the knock-in efficiency (Fig. 5g). Together, these results indicate that the ATM–p53–caspase pathway mediates the cytotoxic response induced by AAV donors. Consistent with this mechanism, both AAV-induced phosphorylation of p53 and activation of caspase 3 were attenuated in ATM-knockout reporter ES cells compared with wild-type cells (Supplementary Fig. 8).

To clarify why suppression of the apoptotic pathway increased the knock-in efficiency, we evaluated the relative copy number of donor vectors in cells using fluorescence intensity as an indicator. Because of the size limitation of DNA that can be packaged into AAV vectors, we first used a plasmid donor with HITI (pReporter–donor–HITI) instead of an AAV donor vector (Fig. 6a). To use blue fluorescent protein (TagBFP) as a

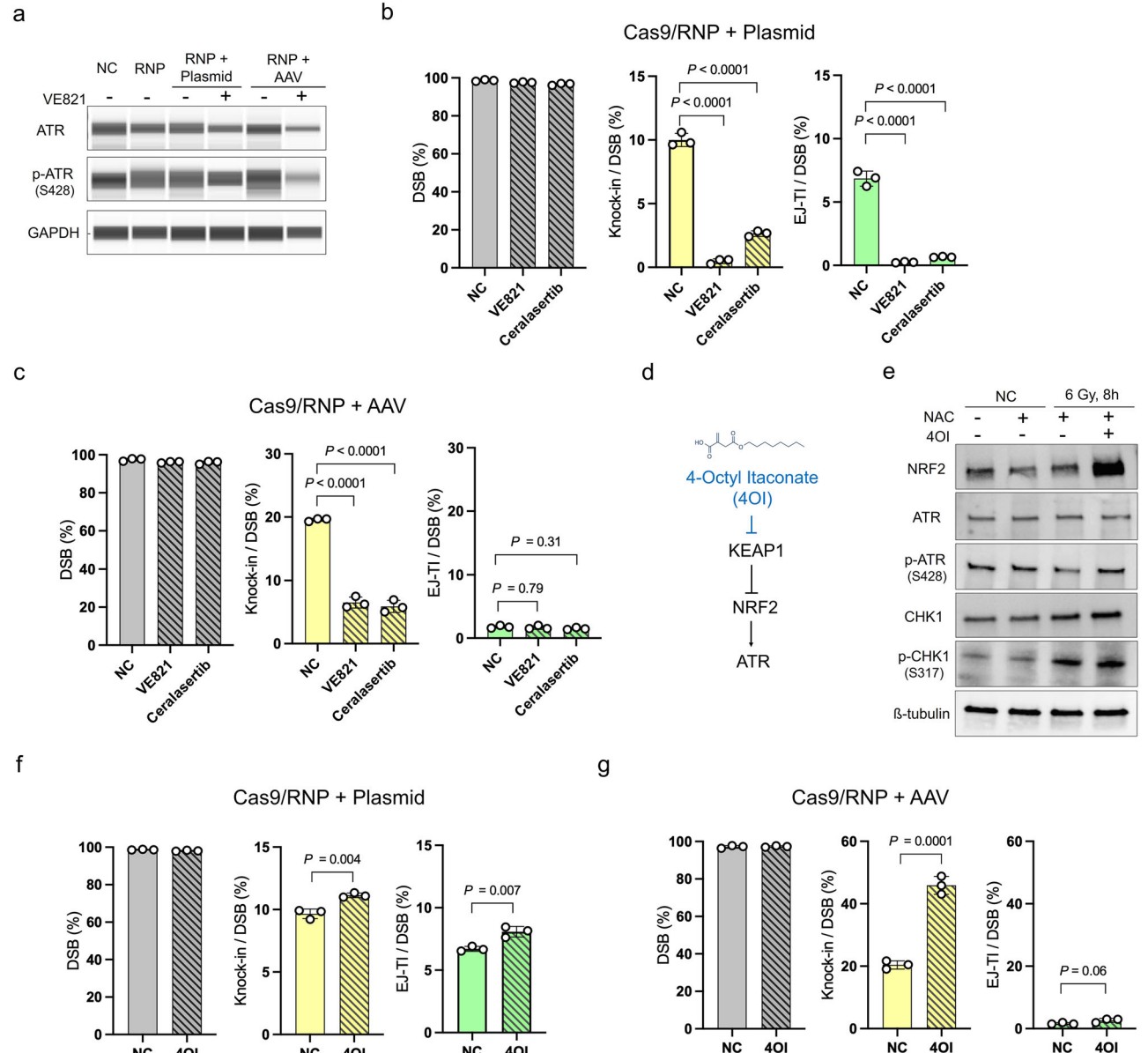

**Fig. 2 | ATR is essential for knock-in, regardless of the donor vector type, and its activation enhances the knock-in efficiency. a** Western blotting illustrating that the ATR inhibitor (10 µM VE821) effectively inhibits ATR activity in genome-edited cells. Protein was harvested 12 h post-genome editing. **b** Genome-editing results with a plasmid donor vector following treatment with two ATR inhibitors (10 µM VE821 and 1 µM ceralasertib). The ATR inhibitors significantly decreased the efficiencies of both knock-in and EJ-TI, underscoring the role of ATR in targeted insertion with the plasmid donor vector. Data are presented as mean ± SD. ($n = 3$ independent experiments). Dunnett's multiple comparison tests were conducted. **c** Genome editing results with an AAV donor vector following treatment with ATR inhibitors. The decreased efficiencies of the knock-in underscore the role of ATR in targeted insertion with the AAV donor vector. Data are presented as mean ± SD. ($n = 3$ independent experiments). Dunnett's multiple comparison tests were conducted. **d** Schematic representation of the hypothesis to modulate ATR activity

through the KEAP1 inhibitor. **e** Western blotting illustrating the upregulation of NRF2 following treatment with a KEAP1 inhibitor (300 µM 4-Octyl Itaconate (4OI)) in cells with IR-induced (6 Gy) DNA damage. A modest increase in phosphorylated ATR was observed. To prevent IR-induced oxidation, the cells were treated with 5 mM N-acetyl-L-cysteine (NAC). Protein extraction was performed 8 h after exposure. **f** Genome editing results with a plasmid donor vector following treatment with a KEAP1 inhibitor (300 µM 4OI) in reporter ES cells. The KEAP1 inhibitor increased the efficiencies of knock-in and EJ-TI. Data are presented as mean ± SD. ($n = 3$ independent experiments). Unpaired $t$-tests were conducted. **g** Genome editing results with an AAV donor vector following treatment with a KEAP1 inhibitor (300 µM 4OI) in reporter ES cells. The KEAP1 inhibitor significantly increased the knock-in efficiency. Data are presented as mean ± SD. ($n = 3$ independent replicates). Unpaired $t$-tests were conducted.

reporter of the donor-vector copy number, we used TI reporter cells (Fig. 1e) lacking the DSB reporter. Two days after transfection with Cas9/RNP and pReporter–donor–HITI containing CAG-TagBFP, both the knock-in and EJ-TI efficiencies were significantly higher in TagBFP-positive cells compared with TagBFP-negative cells (Fig. 6b), indicating that cells harboring higher copy numbers of donor vectors exhibit increased targeted insertion efficiencies. Given this observation, we next tested whether ATM inhibition

affects the survival of high-copy-number cells. Treatment with an ATM inhibitor increased the proportion of TagBFP-positive cells (Fig. 6c), suggesting that ATM inhibition rescues high-copy-number cells from apoptosis and thereby improves overall knock-in efficiency. We then extended our analysis to AAV-donor conditions by replacing the plasmid-HITI with AAV vectors. As shown in Fig. 6d, increasing the concentration of AAV vectors led to a dose-dependent increase in phosphorylated ATM levels.

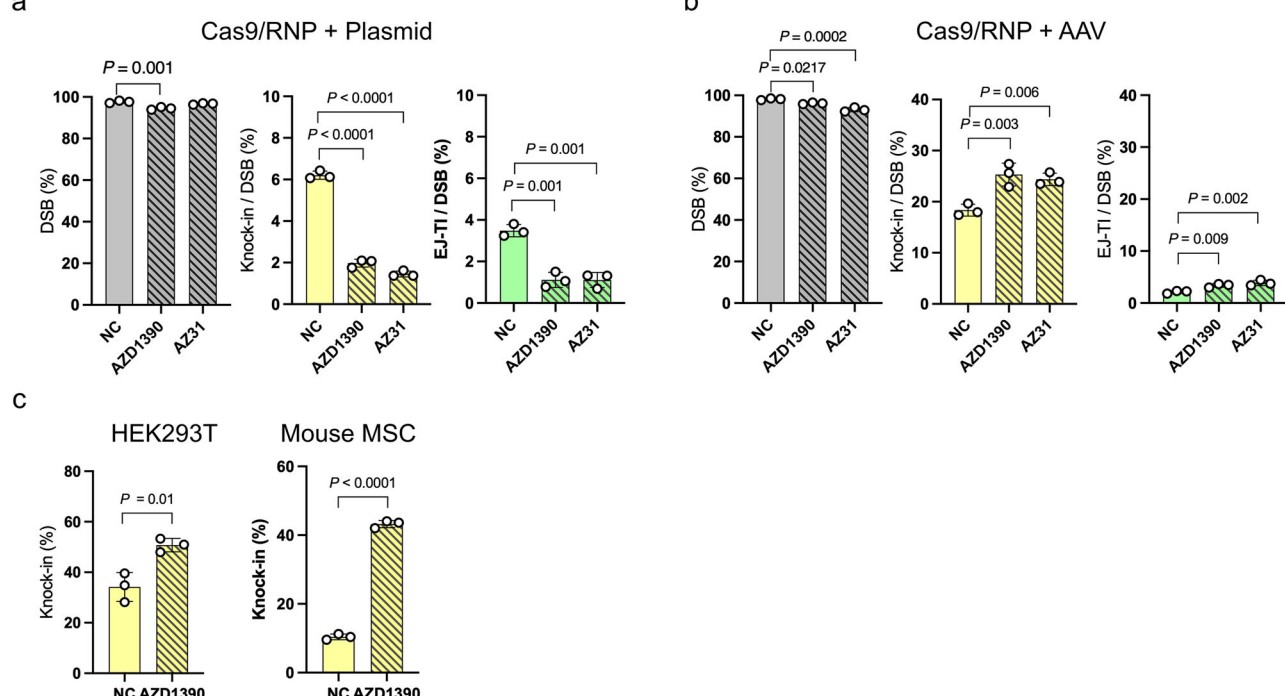

**Fig. 3 | ATM inhibition decreases the knock-in efficiency with circular donor DNA but increases it with linear donor DNA. a** Genome editing results using a plasmid donor vector following treatment with two ATM inhibitors (10 nM AZD1390 and 1 μM AZ31). ATM inhibition significantly decreased the knock-in and EJ-TI efficiencies. Data are presented as mean ± SD. (n = 3 independent replicates). Dunnett's multiple comparison test was conducted. **b** Genome editing results using an AAV donor vector following treatment with two ATM inhibitors (10 nM AZD1390 and 1 μM AZ31). ATM inhibition significantly increased the efficiencies

of both knock-in and EJ-TI. Data are presented as mean ± SD. (n = 3 independent replicates). Dunnett's multiple comparison tests were conducted. **c** Genome editing results using AAV donor vector following treatment with ATM inhibitor (10 nM AZD1390) in HEK293T cells and mouse MSC. In contrast to the triple-reporter ES cells, these experiments were performed using AAV donors designed to fuse EGFP to the C-terminus of the endogenous ACTB: only cells with a successful knock-in at ACTB express EGFP. Data are presented as mean ± SD. (n = 3 independent replicates). Unpaired t-tests were conducted.

This result indicates that cells receiving a high amount of AAV undergo excessive ATM activation, which in turn triggers the p53–caspase 3 apoptotic pathway. Consistent with this mechanism, ATM inhibition significantly increased the intracellular AAV copy number (Fig. 6e), reflecting the survival of cells harboring high AAV copy numbers that would otherwise undergo ATM-dependent apoptosis.

Furthermore, ATM inhibition resulted in a slight but statistically significant reduction in DSB frequency (93.1–96.2% vs. 98.1%, Fig. 3b). The analysis of Supplementary Fig. 6 indicates that c-NHEJ predominantly mediates the repair of the DSB reporter. Based on these observations, we hypothesized that ATM inhibition suppresses the c-NHEJ pathway. To test this idea, we evaluated the activation of DNA-PK, which is a critical component of c-NHEJ. Western blot analysis showed that the level of phosphorylated DNA-PKcs was significantly elevated following the introduction of Cas9/RNP and AAV, but was reduced when cells were treated with an ATM inhibitor (Fig. 7). These findings suggest that the enhancement in the knock-in efficiency observed with ATM inhibition in the presence of an AAV donor vector may result not only from the suppression of apoptosis but also from the partial inhibition of the c-NHEJ pathway (Fig. 8).

## Discussion

In this study, we investigated the roles of ATM in targeted insertion using plasmid and AAV donor vectors. By employing a triple-reporter system that simultaneously detects DSB induction, knock-in, and EJ-TI events, we demonstrated that ATM exhibits a donor-dependent role. Although ATM is required for efficient targeted insertion, its overactivation in response to AAV donor vectors activates the p53- and caspase 3-mediated apoptotic pathway, thereby reducing the knock-in efficiency. These findings underscore the importance of optimizing genome editing strategies by considering the structural properties of donor DNA.

When plasmid donors were used, ATM inhibition significantly reduced both the knock-in and EJ-TI efficiencies. This finding aligns with ATM's well-established role in DSB repair, facilitating homology-directed repair through phosphorylation of key recombination factors such as CtIP, BRCA1 and RAD51[25–28]. In contrast, when AAV donors, which exist as linear ssDNA, were used, ATM inhibition improved knock-in efficiencies, revealing a donor-type-dependent effect. We traced this divergence to the structure of the donor DNA. Our results indicate that the circular or linear structure of the donor critically determines its response to ATM inhibitors, regardless of whether the donor is ssDNA or dsDNA (Fig. 4). Although it remains unclear whether ATM directly recognizes DNA ends, our observations suggest that ATM is activated by these exposed termini, because circular plasmid donors induced only a slight increase (Fig. 5a). Mechanistically, numerous DNA ends typically signal extensive genomic fragmentation and consequently activate the p53–caspase-3–mediated apoptotic pathway[29]. In our experiments, cells that received high copy numbers of AAV vectors exhibited dose-dependent increases in ATM phosphorylation (Fig. 6d), which were accompanied by downstream apoptotic signaling (Fig. 5b, c). This suggests that the cells may misinterpret the abundant AAV genomes as extensive DNA damage, thereby triggering an excessive DDR. These findings are consistent with previous reports showing that AAV can induce apoptosis, highlighting the inherent cytotoxicity of its DNA[30,31]. On the other hand, both our plasmid-HITI-based experiment (Fig. 6b) and previous AAV-based study[32] demonstrated that cells with higher copy numbers of donor vectors exhibited increased targeted insertion efficiency. As a result, cells harboring high copy numbers of donor vectors, which are intrinsically more prone to targeted insertion, were preferentially eliminated through ATM-dependent apoptosis, thereby reducing the overall knock-in and EJ-TI efficiencies.

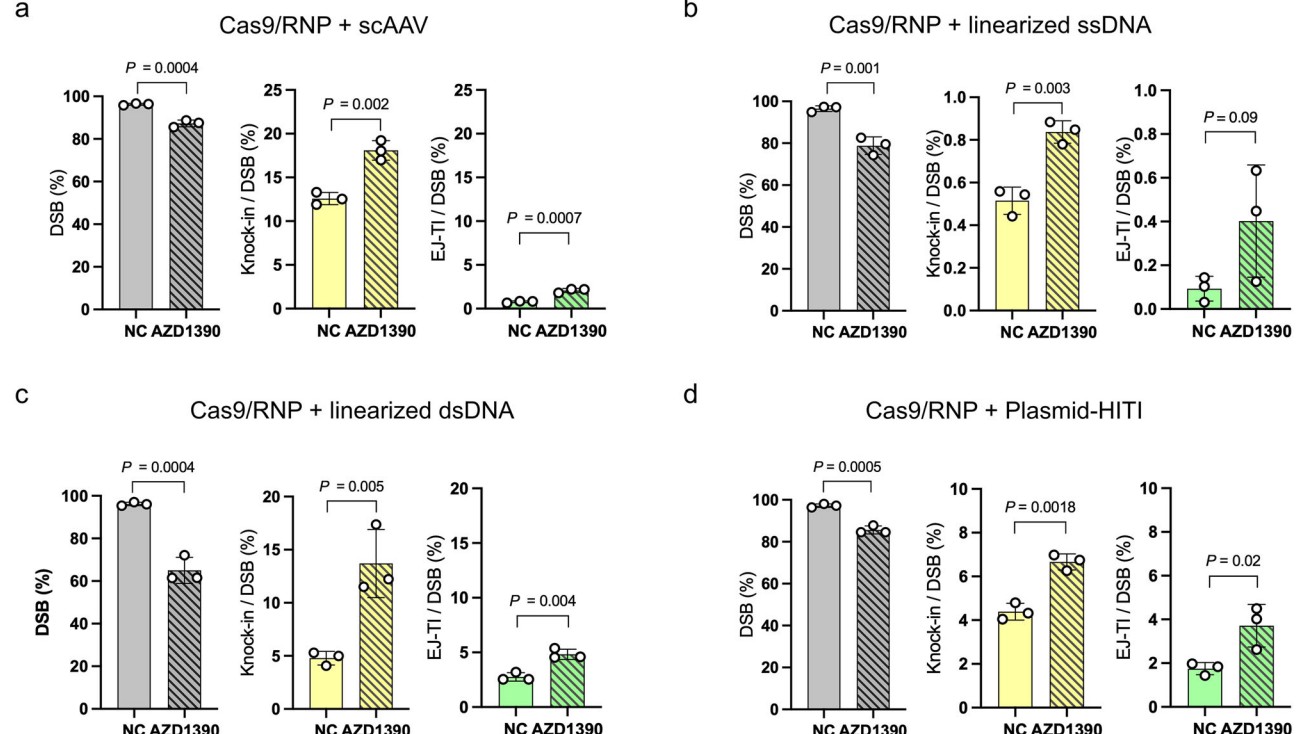

**Fig. 4 | ATM inhibition enhanced the knock-in efficiency with linearized DNA, regardless of the strand type.** Genome editing using **a** self-complementally AAV (scAAV), **b** linearized single-stranded DNA (ssDNA), **c** linearized double-stranded DNA (dsDNA), or **d** plasmid donor vector with HITI following ATM inhibitor treatment (10 nM AZD1390). Data are presented as mean ± SD. ($n = 3$ independent replicates). Unpaired $t$-tests were conducted.

Accordingly, this AAV-induced, ATM-dependent apoptosis provides a mechanistic rationale for why ATM inhibition increases knock-in efficiency. ATM inhibitors effectively suppressed ATM-dependent apoptotic signaling (Fig. 5d–f). As a result, cells harboring high copy numbers of donor DNA, which are inherently predisposed to efficient targeted insertion, survived and became enriched within the population (Fig. 6c, e). In ATM-knockout cells, both AAV-induced phosphorylation of p53 and activation of caspase 3 were attenuated compared with wild-type cells, showing a similar trend to that observed with pharmacologic inhibition, although the extent of attenuation was weaker (Supplementary Fig. 8b, Fig. 5d). This weaker phenotype is likely due to the loss of ATM's scaffold functions, which may permit compensatory rerouting of DNA damage signaling. Supporting this interpretation, a previous study showed that kinase-dead ATM, which retains the scaffold but lacks catalytic activity, exhibits a stronger phenotype than ATM-knockout cells[33]. Taken together with this previous report, these findings collectively indicate that the phenotypes observed with ATM inhibitors reflect the specific inhibition of ATM kinase activity, rather than off-target effects of the compounds. This distinction between pharmacologic inhibition of ATM kinase activity and genetic loss of ATM is important when interpreting prior chromosomal HDR reporter studies lacking abundant linear donor DNA (e.g. I-SceI-induced DR-GFP assays), in which genetic loss of ATM had little impact on HDR[34,35]. By contrast, in our HDR-mediated targeted integration assay using a circular plasmid donor, pharmacologic ATM inhibition markedly reduced knock-in efficiency (Fig. 3a). This apparent discrepancy may reflect fundamental differences between pharmacologic inhibition of ATM kinase activity and genetic loss of ATM, including compensatory rerouting of DNA damage response pathways.

Furthermore, ATM inhibition slightly but significantly decreased the DSB reporter readout (Fig. 3) and reduced DNA-PKcs activity (Fig. 7). Because the DSB reporter is repaired predominantly by c-NHEJ (Supplementary Fig. 6b), these findings suggest that the ATM inhibitor partially attenuated ATM-dependent c-NHEJ signaling via DNA-PK inhibition.

Additionally, ATM has been reported to phosphorylate 53BP1 and promote the recruitment of downstream end-protection factors such as RIF1, which channel DSBs toward c-NHEJ[36,37]. Consistent with these established functions, it is plausible that the ATM inhibitor weakens not only DNA-PKcs activation but also the ATM–53BP1–RIF1 axis, thereby reducing c-NHEJ activity. Such suppression of c-NHEJ may secondarily bias repair toward knock-in under ATM inhibition, in addition to the dominant effect of rescuing cells with high AAV copy numbers from ATM-dependent apoptosis described above. In support of this, prior studies have shown that inhibition of c-NHEJ factors such as 53BP1 and DNA ligase IV increases knock-in efficiency[8,38]. This interpretation also explains the observed increase in EJ-TI (Fig. 3b), which in our system proceeds through c-NHEJ-independent mechanisms (Supplementary Fig. 6b–d). This donor-dependent effect of ATM inhibition is not restricted to mouse ES cells. In the AAV-donor setting, ATM inhibition increased knock-in efficiency in HEK293T and MSCs (Fig. 3c), indicating broad applicability across pluripotent, renal, and mesenchymal lineages.

In contrast to ATM, two chemotype-distinct ATR inhibitors (VE-821 and ceralasertib) concordantly reduced the knock-in and EJ-TI efficiencies regardless of the donor vector type, highlighting ATR's essential role in facilitating targeted insertion (Fig. 2b, c). ATR is activated by ssDNA intermediates generated during DSB resection in the HR pathway, which serve as key signals for recruiting repair proteins and ensuring accurate DSB repair[26,39,40]. In EJ-TI, ATR may contribute indirectly by maintaining genome stability and coordinating the recruitment of end-joining factors. Although the precise mechanisms remain unclear, ATR plays a critical role in EJ-TI as well as knock-in. In this study, we demonstrated that ATR activation via NRF2 stabilization through KEAP1 inhibition improved the knock-in and EJ-TI efficiencies (Fig. 2f, g). This effect arises from the natural regulation of NRF2, which is normally suppressed by KEAP1 via the ubiquitin–proteasome system[41]. These findings highlight the KEAP1-NRF2-ATR pathway as a promising target for increasing knock-in efficiency. Because ATR is essential in mouse ES cells and ATR knockout lines

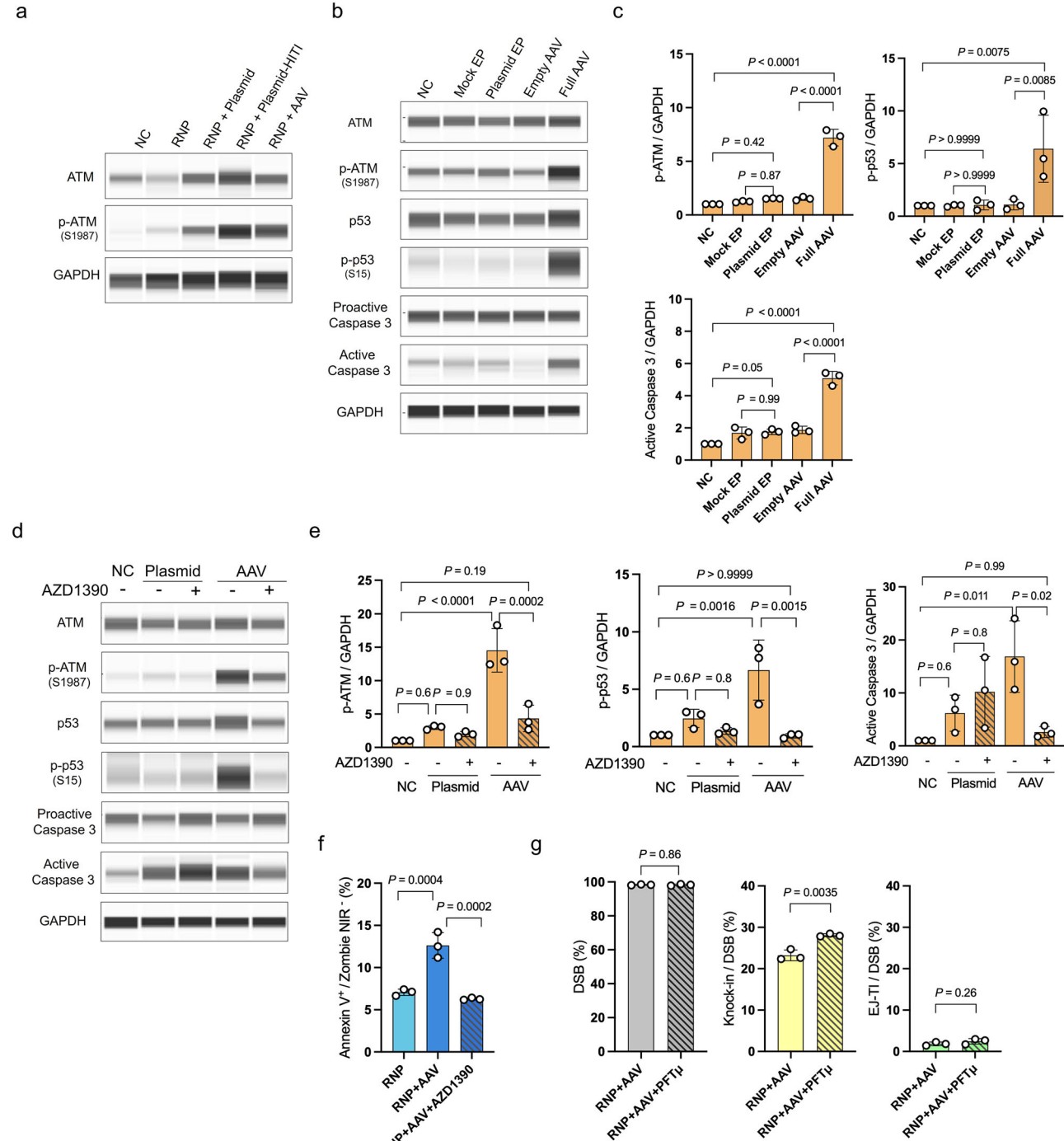

**Fig. 5 | AAV induces the overactivation of ATM and the subsequent activation of the p53-Caspase 3 pathway. a** Western blot analysis showing that ATM was markedly activated during genome editing with an AAV donor or a plasmid donor containing the HITI sequence, whereas only weak activation was observed with a plasmid lacking the HITI sequence. Protein was harvested 4 h after genome editing. **b** Western blot analysis demonstrating that AAV transduction alone activates the ATM-p53-caspase 3 pathway. "EP" stands for electroporation. **c** Quantification of the Western blot results. The signal intensities were normalized to GAPDH. Fold changes are shown relative to the negative control. Data are presented as mean ± SD. (*n* = 3 independent replicates). Tukey's multiple comparison tests were conducted. **d** Western blot analysis showing that treatment with an ATM inhibitor reduces the activation of the ATM-p53-caspase 3 cascade. **e** Quantification of the Western blot

results. The signal intensities were normalized to GAPDH. Fold changes are shown relative to the negative control. Data are presented as mean ± SD. (*n* = 3 independent replicates). Tukey's multiple comparison tests were conducted. **f** Percentage of early apoptotic cells (Annexin V⁺ among Zombie NIR⁻ cells) after genome editing with an AAV donor, with or without ATM inhibitor treatment (10 nM AZD1390). Cells were evaluated 4 h after genome editing. Data are presented as mean ± SD. (*n* = 3 independent replicates). Tukey's multiple comparison tests were conducted. **g** Genome editing results using an AAV donor vector following treatment with a p53 inhibitor (10 μM PFTμ). The knock-in efficiency was significantly increased upon p53 inhibition, highlighting the role of p53 in AAV-mediated knock-in. Data are presented as mean ± SD. (*n* = 3 independent replicates). Unpaired *t*-tests were conducted.

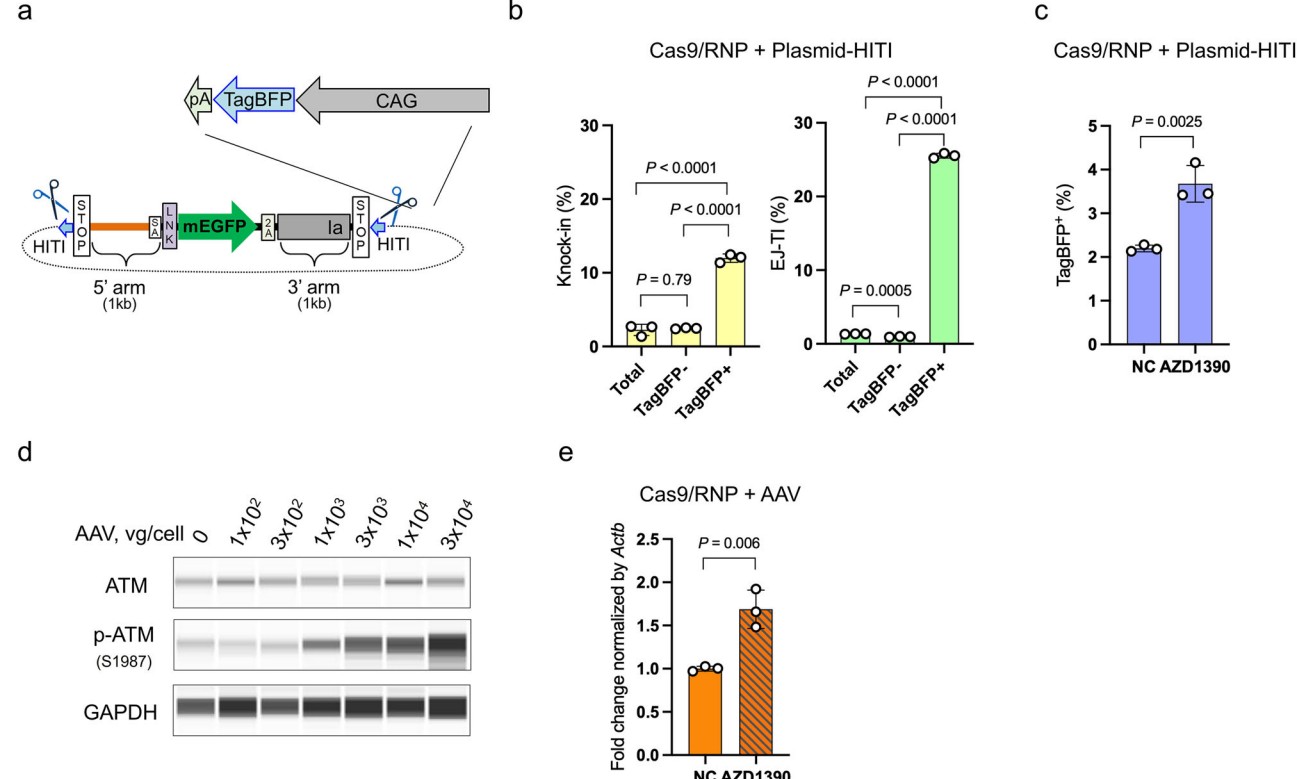

**Fig. 6 | The ATM inhibition increases knock-in efficiency by enhancing survival of cells with high copy number of donor vectors. a** Schematic illustration of the donor vector construct designed to detect donor DNA content in cells. The illustrations of scissors were created with BioRender.com. **b** Genome editing results using the construct from panel (**a**) as a donor vector. TagBFP-positive cells represent those with a higher copy number of donor vectors. The frequency of targeted insertions (knock-in and EJ-TI) was higher in the TagBFP-positive cells than in the TagBFP-negative cells. Data are presented as mean ± SD. (*n* = 3 independent replicates). Dunnett's multiple comparison tests were conducted. **c** Genome editing results following ATM inhibitor treatment (10 nM AZD1390). Treatment with the ATM inhibitor increased the proportion of TagBFP-positive cells, indicating that ATM inhibition enhances the survival of cells with a higher copy number of donor vectors. Data are presented as mean ± SD. (*n* = 3 independent replicates). Unpaired *t*-tests were conducted. **d** Western blot analysis of ATM activation in triple-reporter cells transduced with AAV at increasing concentrations ($1 \times 10^2$ to $3 \times 10^4$ vg/cell). Protein was harvested 4 h post-transduction. Phosphorylated ATM levels increased in a dose-dependent manner. **e** qPCR analysis to determine donor DNA copy number in cells transduced with AAV donor vectors ($1 \times 10^4$ vg/cell) with or without ATM inhibitor (10 nM AZD1390) treatment. Genomic DNA was extracted 4 h after transduction. Data are presented as mean ± SD. (*n* = 3 independent replicates). Unpaired *t*-tests were conducted.

cannot be established[42], genetic reproduction of inhibitor phenotypes is not feasible. Nevertheless, the concordant decreases with VE-821 and cer-alasertib, together with the reciprocal increase upon ATR/CHK1 activation (Fig. 2), provide strong evidence against off-target effects of the ATR inhibitors.

To comprehensively analyze genome editing outcomes, we developed a triple-reporter system that enables the simultaneous detection of DSB induction, knock-in, and EJ-TI. Our TagBFP-based DSB reporter is a mutation-derived proxy rather than a direct physical measurement of DSB formation. It cannot capture precise end-joining events or the C-to-T substitution within the TagBFP chromophore. Nevertheless, the detection probability was very high. Knock-in and EJ-TI were observed only within the TagBFP-negative gate, whereas TagBFP-positive cells showed no detectable targeted insertion (Fig. 1h). In addition, approximately 97% of TagBFP-positive cells retained mCherry expression, and only a minor fraction ( ~ 3%) lacked mCherry expression. These results confirm that the DSB reporter reliably reflects Cas9-induced DSBs. The introduction of the TI reporter, which resulted in heterozygous *Actb* knockout, did not cause any obvious adverse effects, including changes in cell proliferation. This observation is consistent with previous reports showing that heterozygous *Actb* knockout does not affect embryonic development[43]. The proposed system enabled outcome-focused evaluation of targeted insertion. As an initial demonstration of the system's utility, we used a plasmid donor to assess the effects of pharmacologic

modulation of DNA repair pathways on these outcomes. DNA-PK inhibition slightly but significantly reduced the fraction of TagBFP-negative cells, consistent with the dominant contribution of c-NHEJ in DSB repair and suggesting partial compensation by alt-EJ in the DSB reporter. However, the effect on knock-in efficiency was negligible, despite previous reports of improved efficiency following DNA-PK inhibition in certain contexts[10]. Together with the ATM inhibitor data described above, these observations suggest that suppression of c-NHEJ by DNA-PK inhibition alone might be insufficient to substantially enhance knock-in in our system. This limited effect may reflect the high baseline level of homologous recombination activity in pluripotent stem cells[44–46] or possible locus-specific differences in repair pathway usage, which are known to be influenced by Cas9 target residence and gRNA–DNA interactions[10,47]. In contrast, the inhibition of RAD51 significantly decreased the knock-in efficiency, highlighting the reliance on HR-mediated repair. In addition, the reduction of EJ-TI efficiency observed upon PARP1 inhibition suggests that PARP1 activity contributes to this process. RAD51 inhibition also reduced EJ-TI efficiency, indicating that RAD51 contributes to this process alongside PARP1. Although the precise interplay remains unclear, these findings suggest the coexistence of HR- and alt-EJ-related processes, or non-canonical roles for repair factors. Because our main aim was to define the roles of ATR and ATM in knock-in and EJ-TI, we did not explore these pathway relationships in depth. Nevertheless, our triple-reporter system

**Article**

**Fig. 7 | ATM inhibition reduces DNA-PK activity, impairing DSB repair via c-NHEJ. a** Western blotting results showing that treatment with the ATM inhibitor suppressed AAV-induced activation of DNA-PKcs, leading to reduced DSB repair via c-NHEJ. Protein was harvested 2 h after genome editing. **b** Quantification of Western blot results. The signal intensities were normalized to GAPDH. Fold changes are shown relative to the negative control. Data are presented as the mean ± SD. (*n* = 3 independent replicates). Tukey's multiple comparison test was conducted.

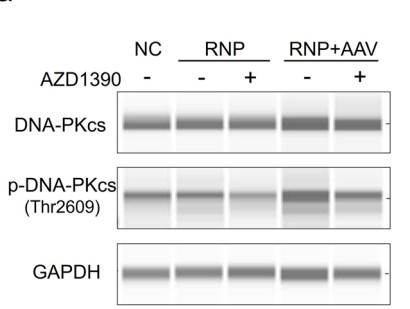

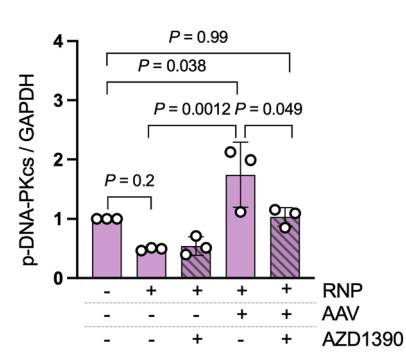

has proven to be instrumental in the simultaneous detection of these key events, making it a powerful tool for conducting more detailed and accurate analyses of genome editing outcomes in future studies.

In summary, the present study revealed that ATR is essential for targeted insertion broadly whereas ATM plays a donor-dependent manner. While ATM is required for efficient targeted insertion, its overactivation in response to AAV donor vectors leads to p53-dependent caspase 3-mediated apoptosis, reducing the efficiency of knock-in. These findings highlight the importance of optimizing genome editing strategies by carefully considering the structural properties of donor DNA and the modulation of DDR pathways.

## Methods
### Cell culture
Mouse ES cells obtained from Professor Takeshi Yagi (Osaka University, Japan) were cultured in 2i medium[48]: Glasgow minimum essential medium (Sigma-Aldrich) containing 10% fetal bovine serum (FBS) (Gibco), 10 ng/mL leukemia inhibition factor (R&D Systems), 3 µM CHIR99021 (StemCell Technologies), 1 µM PD032591 (StemCell Technologies), 0.1 mM beta-mercaptoethanol (Thermo Fisher Scientific), 1× GlutaMAX (Thermo Fisher Scientific), 1 mM sodium pyruvate (Thermo Fisher Scientific), 1× nonessential amino acids (Thermo Fisher Scientific), and 1× penicillin–streptomycin (Thermo Fisher Scientific). The cells were cultured at 37°C in an atmosphere containing 5% CO2 and passaged every 2 days. Mouse bone marrow derived MSC were immortalized by SV40 antigen (addgene). The cells were cultured in Dulbecco's Modified Eagle Medium (DMEM) (Sigma-Aldrich) containing 20% FBS (Corning), 1× GlutaMAX (Thermo Fisher Scientific) and PS (Thermo Fisher Scientific). The cells were cultured at 37°C and 5% CO2 and passaged every 3-4 days. HEK293T cells were cultured in DMEM (Sigma-Aldrich) containing 10% FBS (Corning), 1× GlutaMAX (Thermo Fisher Scientific) and PS (Thermo Fisher Scientific). The cells were cultured at 37°C and 5% CO2 and passaged every 3-4 days. Cells were not routinely tested for mycoplasma contamination, and no additional cell line authentication was performed.

### Plasmid construction
Oligo DNA and primers used for plasmid construction are listed in Supplementary Data 2. pActb–TagBFP: The sequence of TagBFP was ordered as gBlocks from Integrated DNA Technologies (IDT). Then, it was amplified via PCR using a primer set that included a linker sequence directly before TagBFP. The 5′ and 3′ arms were amplified from the mouse genomic DNA via PCR. The three PCR fragments were cloned into linearized pUC19 (Takara Bio) using an NEBuilder cloning kit (NEB).

pActb–lacZ–mCherry: To extend the length of intron 3 of *Actb*, the sequence listed in Supplementary Table 1 was ordered as gBlocks and amplified by PCR. Oligo DNAs were synthesized, including splice acceptor and linker sequences, as well as the recognition sequence for gRNA-cutting TagBFP. The sense and antisense strands were annealed. LacZ and mCherry

were amplified using PCR. The 5′ and 3′ arms were obtained by PCR amplifying mouse genomic DNA. These five fragments were assembled into the PCR-amplified linearized pUC19 with a primer set using the NEBuilder cloning kit.

pReporter–donor: Oligo DNA containing linker sequences and 2 A sequences were synthesized for both sense and antisense strands and annealed. From the pActb–lacZ–mCherry construct, PCR was used to create a 5′ arm containing an upstream STOP sequence and a 3′ arm containing a downstream STOP sequence. mEGFP was amplified using PCR. These five fragments were assembled into the PCR-amplified linearized pUC19 backbone with a primer set using the NEBuilder cloning kit.

pReporter–donor–HITI: Sense and antisense strands of the "homology-independent targeted insertion (HITI)" sequences[22], designed for both the 5′ and 3′ ends, were synthesized, annealed, and inserted into the pReporter–donor vector using an NEBuilder cloning kit. Insertion of the 5′ end was performed between the HindIII and NotI sites located upstream of the pReporter–donor vector's 5' STOP sequence. The 3′ end HITI fragment was inserted between the AflII and SalI sites located downstream of the 3′ STOP sequence.

pAAV–Reporter–donor: pReporter–donor was digested with AflII and HindIII to excise the insert, which was then cloned into a backbone derived from pAAV–CMV–EGFP (Takara Bio).

Cas9/gRNA-expressing vector: To introduce gRNA sequences into the Cas9-expressing vector, pX330-U6-Chimeric_BB-CBh-hSpCas9 (a gift from Feng Zhang, Addgene #42230) was digested with BbsI. Annealed oligonucleotides were subsequently inserted into digested vectors using DNA ligases (Toyobo).

plsODN–Reporter–donor: The plasmid was constructed using pReporter-donor and plsODN3 (BDL).

pAAV-mActb-donor: Oligo DNA containing linker sequences were synthesized for both sense and antisense strands and annealed. EGFP DNA was amplified from pAAV-Reporter-donor, 1 kb homology arm for both 5' end and 3'ends from wild-type mouse cell DNA. These four fragments were assembled into the PCR-amplified linearized pUC19 using the NEBuilder cloning kit. Then, the plasmid was digested with AgeI and BglII to excise the insert which was then cloned into a AAV backbone. For AAV quantifications, the part of SV40 poly A signal was amplified and inserted to the SalI digested plasmid by NEBuilder cloning kit.

pAAV-hACTB-donor: Oligo DNA containing linker sequences were synthesized for both sense and antisense strands and annealed. EGFP DNA was amplified from pAAV-Reporter-donor, 1 kb homology arm for both 5' end and 3'ends from human genomic DNA. A part of SV40 poly A signal was amplified as well. These five fragments were assembled into a backbone derived from pAAV–CMV–EGFP using the NEBuilder cloning kit.

### Establishment of triple-reporter ES cells
Mouse ES cells (1 × 10^6 cells) were suspended in 100 µL of Opti-MEM (Gibco), with either 3 µg of pX330-Actb-Exon6 and 10 µg of pActb–TagBFP

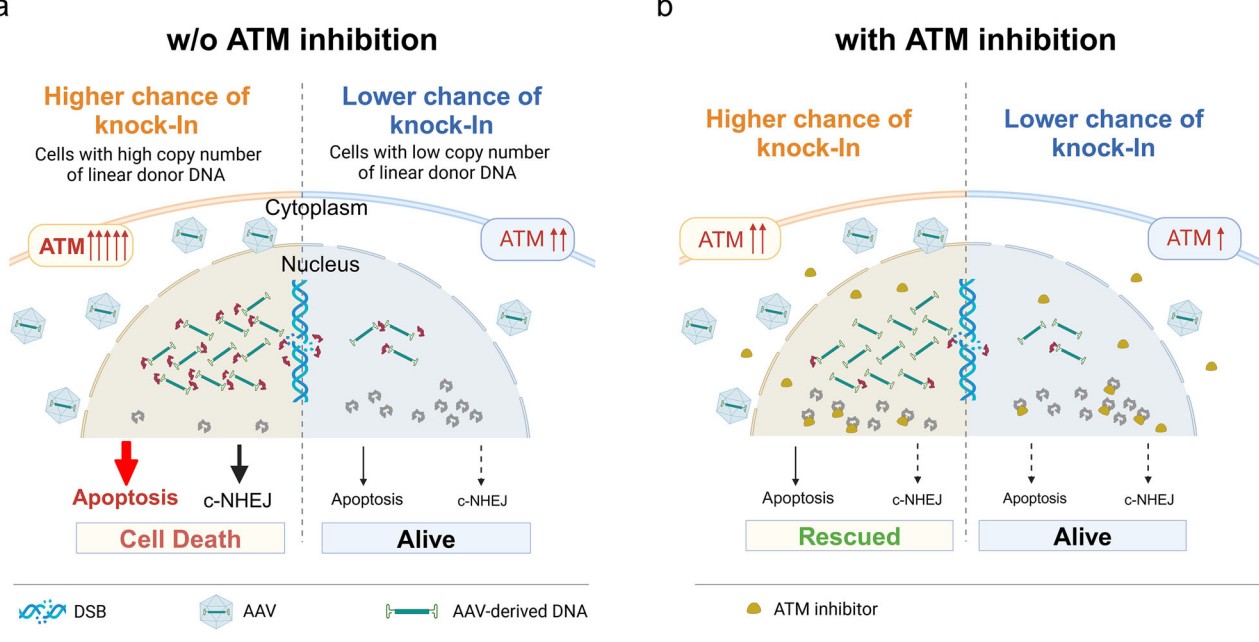

**Fig. 8 | Proposed mechanism by which ATM inhibition enhances the knock-in efficiency with linear donor DNA. a** Without ATM inhibition, cells that take up a high copy number of linear donor DNA (such as AAV-derived DNA) experience overactivation of ATM. This triggers the p53–caspase 3 apoptotic pathway, resulting in the death of cells that would otherwise have a high probability of successful knock-in. Concurrently, the classical non-homologous end-joining (c-NHEJ) remains active, further limiting the homology-directed knock-in events. **b** With ATM inhibition, the overactivation of ATM and its downstream p53–caspase 3 pathway is suppressed, thereby rescuing high-copy-number cells from apoptosis. In addition, partial suppression of c-NHEJ contributes to a higher frequency of knock-in. As a result, the overall knock-in efficiency was significantly increased under ATM-inhibited conditions. Created with BioRender.com.

or 1.5 μg of pX330-Actb-Intron3, 1.5 μg of pX330-Actb-Exon6, and 10 μg of pActb–lacZ–mCherry. Each mixture was transferred to a 2-mm electroporation cuvette and electroporated using NEPA21 (Nepagene). The electroporation settings were set to 145 V for 2 ms with five poring pulses and 20 V for 50 ms with five transfer pulses. After 4 days, TagBFP- or mCherry-positive cells were sorted by single-cell sorting using an SH800 cell sorter (Sony Biotechnology) for cloning. Heterozygous inserted clones were selected using PCR (Supplementary Data 3). The DSB detection ability of the reporter system (Actb–TagBFP) was verified using a Surveyor Mutation Detection Kit (Integrated DNA Technologies) and deep sequencing (Supplementary Data 3).

**Preparation of the Cas9/ribonucleoprotein**
To prepare the Cas9/RNP complex, tracrRNA (IDT) and target-specific crRNA (IDT) were mixed, incubated for 5 min at 95°C, and cooled to room temperature to form a duplex[49]. The target-specific duplex was mixed with Cas9 protein (final: 18.6 μM) (IDT) and incubated for an additional 20 min at room temperature.

**Preparation of AAV, scAAV, ssDNA, and dsDNA donor vectors**
Recombinant AAV6s were purchased from VectorBuilder, which had been purified using a cesium chloride density gradient. scAAV6 was produced using the AAVpro Helper Free System (Takara Bio) according to the minimal purification method[50]. A mixture of the pscAAV–Reporter–donor, pHelper, and pRC6 coding AAV2 rep and AAV6 capsid genes was formulated with a mass ratio of 1:2:2. HEK293T cells were transfected with a mixture of plasmid DNA and PEI Max (Polysciences) at a mass ratio of 1:6. After 24 h, the culture medium was replaced with FBS-free DMEM, and the cells were cultured for 5 days. The medium was harvested and filtered through a 0.22-μm filter (Merck Millipore). The AAV in the filtrate was concentrated using a 100-k molecular weight cut-off Vivaspin 20 column (Sigma-Aldrich) at 2,600 rpm for 3 h. AAV was washed twice in PBS using Vivaspin 20. To determine the concentration of AAV, DNA extraction was performed using the phenol–chloroform–isoamyl alcohol method. qPCR was used to quantify the AAV vector genome (vg) (Supplementary Data 3). To prepare dsDNA, the pReporter–donor plasmid was digested with HindIII and AflII (NEB). The digested DNA was then separated by electrophoresis on a 1% agarose gel. Ethidium bromide staining was used to visualize the DNA bands. The desired bands were excised and purified using the QIAquick Spin Columns (Qiagen). For ssDNA, the plsODN–Reporter–donor plasmid was digested with Nb.BbvCI and EcoRI (NEB). The digested DNA was separated by electrophoresis on a 1.2% agarose gel. Crystal violet staining was used to visualize the DNA bands. The desired band was excised and purified using a Long ssDNA Preparation Kit (BDL).

**Genome editing**
For genome editing in reporter ES cells using a plasmid donor vector, $1 \times 10^6$ reporter ES cells were suspended in 100 μL of Opti-MEM containing either the Cas9/gRNA-expressing vector or the Cas9/RNP complex (final concentration: 0.744 μM), along with 10 μg of pReporter–donor or pReporter–donor–HITI plasmid. Electroporation was carried out under the conditions described above. For genome editing with the AAV donor vector, cells electroporated with the Cas9/RNP complex were mixed with $1 \times 10^4$ vg/cell of AAV6 or $3 \times 10^5$ vg/cell of scAAV6 and plated onto gelatin-coated dishes. The different viral genome doses of AAV6 and scAAV6 were due to differences in purification methods, leading to variations in biological titers. When using linearized DNA as a donor, $1 \times 10^6$ reporter ES cells were suspended in 100 μL of Opti-MEM containing the Cas9/RNP complex and 3 μg of linearized dsDNA or 1.5 μg of ssDNA, followed by electroporation. The quantities of linearized dsDNA and ssDNA were adjusted to match the molar quantity of the circular plasmids (pReporter–Donor and pReporter–Donor–HITI). After electroporation, the cells were cultured in 2i medium containing 10% FBS for 1 day, with a subsequent medium change. For genome editing with AAV donor vectors, cells were initially cultured in serum-free 2i medium[51] for 24 hours to prevent the serum-mediated

inhibition of AAV transduction[52]. The medium was then switched to 2i medium containing 10% FBS. In experiments conducted outside the drug screen, the inhibitors listed in Supplementary Table 2 were applied immediately after electroporation and maintained for 24 hours. Cells were then cultured in inhibitor-free medium for an additional 48 hours, and fluorescence analysis was performed 72 hours after electroporation using an SH800 cell sorter (SONY). To exclude dead cells, 7-AAD (BD Biosciences) staining was performed prior to data collection. Knock-in and EJ-TI efficiencies were normalized to the DSB induction frequency (TagBFP-negative fraction) in each independent experiment to control for variability in delivery and cleavage efficiency.

To perform genome editing in mouse MSC, we targeted the *Actb* locus for promoterless EGFP knock-in. As the donor DNA lacks a promoter, EGFP-positive cells represent successfully knocked-in cells. Cells were electroporated with the Cas9/RNP complex, mixed with $1 \times 10^4$ vg/cell of AAV6, and plated. Electroporation was performed using four poring pulses at 140 V for 2 ms, followed by five transfer pulses at 20 V for 50 ms. After electroporation, cells were cultured in serum-free DMEM for 24 hours, then transferred to DMEM supplemented with 20% FBS for an additional 48 hours.

For genome editing in the HEK293T cell line, EGFP was knocked into the human *ACTB* locus using the same promoterless design. As in the mouse model, EGFP expression served as a marker for successful knock-in. Cells were electroporated with the Cas9/RNP complex, mixed with AAV6 at $1 \times 10^4$ vg/cell, and plated. Electroporation was conducted using four poring pulses at 160 V for 2 ms, followed by five transfer pulses at 20 V for 50 ms. Cells were then cultured in serum-free DMEM for 24 hours, followed by DMEM containing 10% FBS for an additional 48 hours. Editing efficiency was assessed using an SH800 cell sorter, and dead cells were excluded by DAPI staining (Dojindo) prior to data acquisition.

### DDR-inhibitor screen
We performed a DDR-inhibitor screen using the DNA Damage/DNA Repair Compound Library (Selleck; 174 compounds) to identify small molecules that modulate targeted insertion efficiency. Reporter ES cells were electroporated with a Cas9/gRNA-expressing plasmid together with a plasmid donor, or with a Cas9/RNP complex alone when using an AAV donor. Electroporated cells were seeded into 96-well plates at 10,000 cells per well, each well containing a single compound (Supplementary Data 1) and AAV when applicable, as described in the previous section. After 24 h of incubation, the medium was replaced with inhibitor-free medium, and cells were cultured for an additional 48 h. Fluorescence signals were analyzed 72 h after editing using an LSRFortessa X-20 flow cytometer (BD Biosciences) equipped with a sample loader. Knock-in and EJ-TI efficiencies were quantified as percentages of live singlets. The screening was performed in triplicate for the plasmid donor condition and once for the AAV donor condition. Compounds showing changes in knock-in efficiency beyond the mean value of the untreated control ± 3 standard deviations (Average ± 3 SD) were defined as hits.

### Establishment of ATM knockout reporter ES cells
ATM knockout reporter ES cells were generated using a Cas9/RNP complex targeting exon 4 of the *Atm* gene (the gRNA sequence is provided in Supplementary Fig. 8). Reporter ES cells ($2 \times 10^6$) were electroporated with the Cas9/RNP complex under the same conditions used for genome editing described above. After electroporation, cells were cultured for three days and subjected to single-cell sorting using an SH800 cell sorter to establish clonal populations. Genomic DNA from individual clones was analyzed by PCR (primer sequences are listed in Supplementary Data 3), followed by Sanger sequencing to identify InDels that introduce frameshift mutations in the *Atm* locus. Clones harboring biallelic frameshift mutations were further validated by Western blot analysis to confirm the complete loss of ATM protein expression.

### qPCR
For allele-specific validation of reporter ES cells, standard curves were generated using serial dilutions of amplicons containing *Actb* exon 3, *TagBFP*, or *mCherry*. Genomic DNA was extracted from reporter ES cell clones using the DNeasy Blood & Tissue Kit (Qiagen). qPCR reactions were performed using PowerUp SYBR Green Master Mix (Applied Biosystems) and primer sets targeting *Actb* exon 3, *TagBFP*, and *mCherry*. Amplification was carried out under identical cycling conditions for all targets. The relative abundance of each reporter sequence was calculated by interpolation from the standard curves to confirm whether *TagBFP* and *mCherry* were integrated into different alleles.

To quantify donor DNA copy number following ATM inhibitor treatment, reporter ES cells were electroporated with the Cas9/RNP complex and transduced with AAV6 donor vectors at $1 \times 10^4$ vg/cell under the same genome editing conditions used in the knock-in experiments described above. After 4 h, genomic DNA was extracted using the DNeasy Blood & Tissue Kit (Qiagen). qPCR was performed using PowerUp SYBR Green Master Mix (Applied Biosystems) with primers targeting *Actb* intron 1 as a single-copy endogenous reference and *mEGFP*. Donor DNA copy number per genome was calculated using the ΔΔCt method normalized to *Actb* intron 1. Primer sequences are provided in Supplementary Data 3.

### Western blotting
Proteins were harvested from the cell culture plates using RIPA buffer (Nacalai Tesque) or M-PER (Thermo Fisher Scientific), both containing protease inhibitors (Nacalai Tesque) and a proteinase inhibitor cocktail (Abcam), after washing the cells twice with ice-cold PBS. The extracted proteins were diluted in Laemmli buffer (Bio-Rad) containing dithiothreitol (DTT) at a final concentration of 100 mM and boiled at 95 °C for 5 min. The proteins were then stored at −80 °C until Western blot analysis. The protein concentration was determined using a Pierce™ 660 nm Protein Assay (Thermo Fisher Scientific). For Western blot analysis, the protein expression or phospho-protein levels of NRF2, ATR, p-ATR, CHK1, p-CHK1, and ß-tubulin were evaluated using traditional Western blotting methods. Specifically, SDS-PAGE was performed on a 4–15% gradient gel (Bio-Rad), and proteins were transferred onto a 0.2-μm PVDF membrane (Bio-Rad) using the TransBlot Turbo system (Bio-Rad). The membrane was blocked with PVDF blocking solution (Toyobo) for 2 h at room temperature and washed with TBS-T buffer (Takara Bio). Primary antibody incubation was performed either at room temperature for 1 h or at 4 °C overnight, depending on the target protein. After washing with TBS-T, the membrane was incubated with the appropriate secondary antibody at room temperature for 30 min, followed by washing with TBS-T at room temperature. Protein signals were detected using the LAS3000-mini (Fujifilm) with a chemiluminescent horseradish peroxidase substrate (Immobilon). For proteins including ATM, p-ATM, p53, p-p53, caspase 3, cleaved caspase 3, DNA-PK, p-DNA-PK, ATR, p-ATR, and GAPDH, capillary Western blotting was performed using the JESS system (Simple Western, Bio-techne) according to the manufacturer's protocol. Protein quantification was performed using Compass SW software and ImageJ. A complete list of the antibodies, reagents, and consumables used in Western blotting is provided in Table 1 and Supplementary Table 3.

### Apoptosis assay
Four hours after genome editing, triple-reporter ES cells ($2 \times 10^5$ cells) were stained with 200 μL of Zombie NIR (BioLegend, 1:400 in PBS) for 15 min at room temperature, followed by incubation in 200 μL of Annexin V Binding Buffer (BioLegend) containing Alexa Fluor 647–Annexin V (BioLegend, 10 μL) for 15 min at room temperature. Cells were immediately analyzed using an SH800 cell sorter. Early apoptotic cells were defined as Annexin V⁺/Zombie NIR⁻.

### Statistics and reproducibility
The statistical analysis was performed using Prism version 10 (GraphPad Software). Data from all experiments are presented as the mean ± standard

**Table 1 | The list of the antibodies**

| No | Name of the antibody | Manufacturer | Cat# | Dilution | |
|---|---|---|---|---|---|
| | | | | Jess | WB |
| 1 | ATR (E1S3S) Rabbit mAb | Cell signaling | 1394 | 1:100 | 1:1000 |
| 2 | Phospho-ATR (Ser428) Antibody | Cell signaling | 2853 | 1:50 | 1:1000 |
| 3 | ATM (D2E2) Rabbit mAb | Cell signaling | 2873 | 1:100 | - |
| 4 | Anti-ATM (phospho S1987) | Abcam | ab315019 | 1:200 | - |
| 5 | Chk1 (2G1D5) Mouse mAb | Cell signaling | 2360 | - | 1:1000 |
| 6 | Human/Mouse/Rat Phospho-Chk1 (S317) Antibody | RnD systems | AF2054 | - | 1:1000 |
| 7 | NRF2 (D1Z9C) XP® Rabbit mAb | Cell signaling | 12721 | - | 1:1000 |
| 8 | p53 (1C12) Mouse mAb | Cell signaling | 2524 | 1:100 | - |
| 9 | Phospho-p53 (Ser15) Antibody | Cell signaling | 9284 | 1:100 | - |
| 10 | Caspase-3 Antibody | Cell signaling | 9662 | 1:100 | - |
| 11 | Cleaved Caspase-3 (Asp175) Antibody | Cell signaling | 9661 | 1:20 | - |
| 12 | DNA-PKcs Polyclonal antibody | Proteintech | 28534-1-AP | 1:100 | - |
| 13 | Phospho-DNA-PK (Thr2609) Polyclonal Antibody | Invitrogen | PA5-105749 | 1:50 | - |
| 14 | β-Tubulin Antibody | Cell signaling | 2146 | 1:100 | 1:1000 |
| 15 | GAPDH Antibody (6C5) | Santa Cruz | sc-32233 | 1:100 | 1:2500 |
| 16 | Peroxidase AffiniPure Donkey Anti-Mouse IgG (H + L) | Jackson Immuno Research Laboratories, Inc. | 715-035-150 | - | 1:5000 |
| 17 | Peroxidase AffiniPure Donkey Anti- Rabbit IgG (H + L) | Jackson Immuno Research Laboratories, Inc. | 715-035-152 | - | 1:5000 |

deviation (SD). The percentage data were arcsine-transformed prior to analysis, and the distribution was assessed using the Shapiro–Wilk test. Values of $p < 0.05$ were considered statistically significant in the unpaired $t$-test, Dunnett's multiple comparison test, and Tukey's multiple comparison test. Parametric and nonparametric tests were selected based on the results of the normality test.

### Reporting summary

Further information on research design is available in the Nature Portfolio Reporting Summary linked to this article.

## Data availability

The raw deep-sequencing reads generated in this study have been deposited in the NCBI Sequence Read Archive (PRJNA1390995). All other data supporting the findings of this study have been deposited in Zenodo (DOI: 10.5281/zenodo.17982883). The gating strategies for all flow cytometry plots are provided in the Supplementary Figs. 9-12. Uncropped PCR and western images are provided as Supplementary Figs. 13-26. All plasmids newly generated in this study have been deposited in Addgene (251409–251418). All other materials are available from the corresponding author upon reasonable request.

## Code availability

This study did not generate or use any original code.

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

## Acknowledgements

The authors acknowledge Hiroko Hayakawa for her expert technical assistance at the Center for Cytometry Research, Jichi Medical University, in operating the LSRFortessa X-20 flow cytometer. We thank Dr. Takeshi Tokuyama for his invaluable technical assistance with Western blotting. In addition, we are grateful to Professor Takeshi Yagi for generously providing the mouse ES cells used to establish the triple-reporter cells. This research was supported by the Japan Agency for Medical Research and Development (AMED) [JP18am0301002 to Y.H., JP22ae0201007 to Y.H., JP22bm0804018 to H.U., JP23bm1123020 to F.N. and HH]. Y.H. and H.H. have received a research fund from Sumitomo Pharma Co., Ltd. as a result of the Collaborative Research Agreement between Jichi Medical University and Sumitomo Pharma Co., Ltd.

## Author contributions

Conceptualization, H.H. and Y.H.; Methodology, H.H.; Investigation, M.N. and H.H.; Writing–Original Draft, M.N. and H.H.; Writing–Review & Editing, H.U., F.N., M.I., and Y.H.; Analysis, M.N., H.H.

## Competing interests

The authors declare the following competing interests: Y.H., M.I., and H.H. are affiliated with a joint research laboratory between Jichi Medical University and Sumitomo Pharma Co., Ltd. A patent application related to

the methods and findings reported in this paper is pending. M.I. is an employee of Sumitomo Pharma Co., Ltd. and Racthera Co., Ltd. All other authors declare no competing interests.
