## [Transparent Peer Review file · Communications Biology]

ATM Inhibition Enhances Knock-in Efficiency by Suppressing AAV-Induced Activation of Apoptotic Pathways

Corresponding Author: Dr Hiromasa Hara

Parts of this Peer Review File have been redacted as indicated to maintain patient confidentiality.

Version 0:

Reviewer comments:

Reviewer #1

(Remarks to the Author)

In this manuscript, the authors developed a triple-reporter system in mouse ES cells that allows simultaneous detection of indel-inducing NHEJ (referred to by the authors as “DSB induction”), HDR-based knock-in, and NHEJ-based targeted integration (EJ-TI) at a defined site of the reporter system. Using this system, they show that ATM inhibition reduces knock-in and EJ-TI efficiencies when circular plasmid donors are used, but increases these efficiencies when linear AAV donors are applied. Further investigation reveals that the linear structure of the AAV donors (and linearized plasmid donors) overactivates the ATM-p53-caspase 3 apoptotic pathway. Based on this, the authors propose a mechanistic link whereby ATM inhibition enhances knock-in and EJ-TI efficiencies by mitigating apoptosis triggered by donor DNA-induced ATM-p53-caspase 3 signaling. This may present a strategy to improve knock-in and EJ-TI outcomes of genome editing. The findings are intriguing and the design of the triple-reporter system is elegant. However, several concerns remain as stated below regarding experimental design, interpretation, and data presentation.

Major comments

- 1) The authors used PCR to confirm targeted integration of the reporters into the *Actb* locus. However, this approach does not exclude the possibility of additional, randomly integrated copies of the reporters, which may confound analysis of NHEJ, HDR, and EJ-TI. More rigorous validation methods such as Southern blotting, digital droplet PCR (ddPCR), or quantitative PCR are recommended to verify single-copy, locus-specific integration.
- 2) In this manuscript, ATM inhibition has been shown to suppress c-NHEJ. Given that EJ-TI is largely mediated by c-NHEJ, which is suppressed by ATM inhibition, how does ATM inhibition paradoxically enhance EJ-TI?
- 3) Page 3 line 68-69: “lack the ability to reliably distinguish precise knock-in events from error-prone end-joining-mediated targeted insertions (EJ-TI).” However, as demonstrated in their own targeted integration assay, junction-specific PCR readily differentiates these events. This statement should be revised for accuracy.
- 4) Fig. 1e and page 4 line 133-134: “In contrast, only mEGFP was expressed in EJ-TI because the splice acceptor was located immediately upstream of mEGFP”. However, there are both a splice acceptor and a stop codon upstream of mEGFP, which should preclude EGFP translation. Why, then, are EJ-TI cells EGFP-positive?
- 5) Also in Fig. 1e, the EJ-TI donor contains homology arms for both sides of the DSB site at the genomic target. This could allow integration via single-strand annealing (SSA) or microhomology-mediated end joining (MMEJ), in addition to HDR or c-NHEJ. Consequently, EJ-TI outcomes may include mEGFP-only, mCherry-only, or double-positive cells. The manuscript does not distinguish between these possibilities, which complicates interpretation of the EJ-TI data.
- 6) The DSB reporter infers DSB induction from indel frequency. However, it is well-established that initial NHEJ repair of Cas9-induced DSBs is typically accurate, and the observed indels largely result from repeated Cas9 cleavage. Further considered the effect of Cas9 target residency besides Cas9 recleavage on indel formation, the reporter may not reliably measure DSB induction as claimed. The observation that a small fraction (~3%) of BFP-positive cells are RFP-negative also raises questions about this interpretation.
- 7) It is well documented that DNA-PKcs inhibition reduces c-NHEJ and enhances HDR. This manuscript reports only a mild decrease in indels and negligible effects on HDR or EJ-TI (Fig. S5b). This unexpected result may be influenced by Cas9 post-cleavage target residency, which affects DSB repair pathway choices (PMID: 35915475). This issue should be further

addressed or discussed.

8) In Fig. 3, linear plasmids should also be included. By directly comparing AAV transduction and circular plasmid electroporation, the authors introduce confounding variables, including differences in delivery method and AAV-specific cellular response, which might contribute to ATM overactivation.

9) In Fig. 4b, TagBFP expression may reflect transfection efficiency more than actual donor copy number. Additional methods are needed to quantify donor DNA copy number more accurately. Moreover, the claim that ATM inhibition increases knock-in by rescuing high-donor-copy cells is based solely on p53 S15 phosphorylation and Caspase-3 activation. Direct measurements of cell viability or apoptosis (e.g., Annexin V/PI staining, live/dead cell assays) would strengthen this conclusion.

10) Earlier studies indicate that ATM is dispensable for HDR in mouse ES cells (e.g., PMID: 23355489, 28659469), yet this manuscript reports a strong HDR enhancement with ATM inhibition. The authors should reconcile these contrasting results in the Discussion.

11) In the section of Statistical analysis, it is not clear how many independent experiments have been done and whether each independent experiment is performed in triplicates. Does "n=3" indicated in the manuscript represent 3 independent experiments, each in triplicates, or one single independent experiment in triplicates? In addition, are the efficiencies of NHEJ, HDR-mediated KI, EJ-TI presented in Figures normalized with transfection efficiencies? This information should also be provided along with transfection efficiency values.

12) To solidify the conclusion that ATM inhibition drives the effects as observed in this manuscript, validation with ATM knockout cells would be highly informative and strengthen the study.

13) Minor comments: In Fig. 1b, the "LNK" element in the reporter should be explained; In Fig 1c, consider showing the full mutation spectrum for better clarity; It is unclear in the manuscript when the inhibitors were added during genome editing assays.

Reviewer #2

(Remarks to the Author)

I co-reviewed this manuscript with one of the reviewers who provided the listed reports. This is part of the Communications Biology initiative to facilitate training in peer review and to provide appropriate recognition for Early Career Researchers who co-review manuscripts.

Reviewer #3

(Remarks to the Author)

The manuscript by Hanazono and colleagues aims to understand the molecular mechanisms of AAV-mediated knock-in (as a donor vector in CRISPR Cas-9 system) in order to evaluate its editing efficacy and improve its therapeutic potential. The authors have developed a novel triple-reporter system that monitors DSB-induction, knock-in, and EJ-TI simultaneously. Using this reporter, they perform a chemical screen to identify regulators of knock-in efficiency. They reveal that ATM inhibition diminishes knock-in efficacy in plasmid donors yet enhances efficiency in AAV donors. Overall, this is a strong and informative manuscript, which will be of interest to the field. However, the authors make some generalizations, using donor vectors to show that increased copy-number increases insertion efficacy, applying this finding to AAV vectors. This, and additional comments are listed below:

1. The authors convincingly show that inhibition of ATM, and subsequent downregulation of apoptosis, is specific to increasing knock-in efficacy in AAV vectors. However, in figure 4, they use donor vectors to show that increased copy-number increases insertion efficacy, applying this finding to AAV vectors. The authors should perform additional experiments using AAV vectors, to investigate if increased copy number directly decreases targeted insertion via activation of ATM and the apoptotic pathway.
2. The authors could explore why RAD51 inhibition promotes an observed decrease in EJ-TI events (as shown in supplementary figure 5) and whether this phenotype is epistatic with the inhibition of PARP1.
3. One confusing aspect of the manuscript is that the results showing that inhibition of ATM in cells harboring donor vectors enhances survival and knock-in efficacy seems to conflict with earlier findings in the paper showing that activation of ATM improves plasmid-mediated knock-in efficacy.
4. Better integration of the supplementary material with the main figures and text would make the paper flow better and easier to follow as a reader. For example, the paper would likely be easier to read if some of the supplementary material was integrated into the main figures rather than stand-alone panels in the supplement.
5. The authors indicate that they performed a "comprehensive drug screening" but very few details are presented. It is only in the Methods section that they indicate that the compound library used was the DNA damage/repair library. This is critical information to be included in the Results section, otherwise it gives the impression that the authors screened a broader library. In addition, the Methods section should include a separate Compound screening section, where details about the screen should be presented (duration of treatment, concentration, number of compounds, number of sorted cells, number of replicates of the screen etc). The current paragraph on page 17 is very brief and it is unclear how the screen was done and how robust the assay was.
6. The screen results presented in Suppl Fig 6 show a number of additional hits (#49 increases and #144 and #151 decrease the signal in 6a; #54 increases and #33 and #144 decrease the signal in 6b). The authors do not comment on this at all. What are these other hits, and were they validated? The manuscript is narrowly focused on ATM and ATR. Expanding to these other hits could be very powerful.
7. Were ATR inhibitors present in the library (one would assume so), and were they identified as hits?
8. A significance weakness of the manuscript is that the authors used only small molecule inhibitors to probe the roles of

ATM and ATR in these processes. Since these chemicals may have off target effects, it is important for the authors to validate some of their main findings using genetic approaches (gene depletion) to confirm the phenotypes observed using the inhibitors.

Reviewer #4

(Remarks to the Author)

I co-reviewed this manuscript with one of the reviewers who provided the listed reports. This is part of the Communications Biology initiative to facilitate training in peer review and to provide appropriate recognition for Early Career Researchers who co-review manuscripts.

Version 1:

Reviewer comments:

Reviewer #1

(Remarks to the Author)

The authors have adequately addressed almost all of my questions with the revision. One minor comment: the statement from Line 364-366, "By contrast, under conditions lacking abundant linear donor DNA (e.g. chromosomal DR-GFP assays), where ATM is not overactivated, ATM inhibition decreases HR37,38 ", is not fully accurate. In these two cited articles, ATM loss does not affect HDR in mouse ES cells; however, treatment with an ATM inhibitor reduces HDR in the work by Chen et al., who speculate that inactivated ATM kinase by the inhibitor in ATM wild-type mouse ES cells might interfere recruitment of HDR factors to the I-SceI-induced DSBs. This seems to have nothing to do with ATM overactivation.

Reviewer #2

(Remarks to the Author)

I co-reviewed this manuscript with one of the reviewers who provided the listed reports. This is part of the Communications Biology initiative to facilitate training in peer review and to provide appropriate recognition for Early Career Researchers who co-review manuscripts.

Reviewer #3

(Remarks to the Author)

I commend the authors for an excellent job at addressing the reviewers' comments, and thus enhancing the strengths of the manuscript. In my opinion, the manuscript is acceptable for publication.

Reviewer #4

(Remarks to the Author)

I co-reviewed this manuscript with one of the reviewers who provided the listed reports. This is part of the Communications Biology initiative to facilitate training in peer review and to provide appropriate recognition for Early Career Researchers who co-review manuscripts.

Response to Reviewer #1

We sincerely thank Reviewer #1 for the insightful and constructive comments. The detailed feedback has been extremely valuable in improving the clarity and rigor of our manuscript.

We have carefully addressed each point as outlined below.

Major comments

- 1) The authors used PCR to confirm targeted integration of the reporters into the *Actb* locus.

However, this approach does not exclude the possibility of additional, randomly integrated copies of the reporters, which may confound analysis of NHEJ, HDR, and EJ-TI.

More rigorous validation methods such as Southern blotting, digital droplet PCR (ddPCR), or quantitative PCR are recommended to verify single-copy, locus-specific integration.

We performed qPCR analysis to determine the copy number of both the DSB reporter (*Actb*-TagBFP) and TI reporter (*Actb*-lacZ-mCherry) in the established cell lines. The results showed that each reporter was integrated as a single copy per diploid genome (Supplementary Fig. 5b). Combined with the PCR data presented in the original submission (Supplementary Fig. 5a in the revised manuscript), these results confirm specific integration of the reporter cassettes into the *Actb* locus without additional random insertion.

2) In this manuscript, ATM inhibition has been shown to suppress c-NHEJ. Given that EJ-TI is largely mediated by c-NHEJ, which is suppressed by ATM inhibition, how does ATM inhibition paradoxically enhance EJ-TI?

As shown in Supplementary Fig. 6, while DNA-PK inhibition did not suppress EJ-TI, inhibition of RAD51 and PARP1 reduced EJ-TI. These results indicate that EJ-TI process is independent of c-NHEJ and instead involves the coexistence of HR- and alt-EJ-related mechanisms or noncanonical roles of repair factors.

Moreover, ATM inhibition prevents AAV-induced p53–caspase-3–mediated apoptosis, allowing high-donor-copy cells—which inherently show higher probabilities of both knock-in and EJ-TI—to survive (Fig. 5, 6). This represents a distinct, apoptosis-related mechanism rather than a direct modulation of DNA repair pathways.

Therefore, the finding that ATM inhibition, which also reduces c-NHEJ activity, enhances EJ-TI efficiency is entirely consistent with these observations and not paradoxical. We have clarified this point in the Discussion (page 12, lines 329–332; page 14, lines 409–416).

- 3) Page 3 line 68-69: "lack the ability to reliably distinguish precise knock-in events from error-prone end-joining-mediated targeted insertions (EJ-TI)." However, as demonstrated in their own targeted integration assay, junction-specific PCR readily differentiates these events. This statement should be revised for accuracy.

We agree that junction-specific PCR can qualitatively distinguish knock-in events from EJ-TI. To clarify this point and avoid overstatement, we have revised the sentence as follows:

Page 3, Line 66-70: Conventional approaches, such as PCR-based assays¹⁷⁻¹⁹, which are subject to amplification biases, and fluorescent reporter systems^{20,21}, which exhibit limited ability to reliably and quantitatively distinguish precise knock-in events from error-prone end-joining-mediated targeted insertions (EJ-TI), have inherent methodological limitations.

- 4) Fig. 1e and page 4 line 133-134: "In contrast, only mEGFP was expressed in EJ-TI because the splice acceptor was located immediately upstream of mEGFP" . However, there are both a splice acceptor and a stop codon upstream of mEGFP, which should preclude EGFP translation. Why, then, are EJ-TI cells EGFP-positive?

When EJ-TI occurs, two splice acceptors (SAs) are available: (i) a genomic SA

embedded in the reporter locus upstream of a stop codon, and (ii) a donor-derived SA positioned immediately upstream of mEGFP within the donor cassette. Both SAs can be used, generating two transcript isoforms.

- Splicing to the genomic SA retains the stop codon and therefore yields a non-productive transcript (no mEGFP)
- Splicing to the donor-derived SA skips the stop codon in the mature transcript and generates an in-frame mEGFP transcript, explaining mEGFP positivity observed in EJ-TI cells.

The coexistence of these two isoforms within the same cells also provides a plausible explanation for the lower mEGFP fluorescence observed in EJ-TI compared with knock-in.

For clarity, we have revised the Results (page 4, lines 104-114) to explicitly state that mEGFP in EJ-TI is generated by splicing to the donor-derived SA, whereas the genomic SA yields a non-productive isoform. Consistent with this mechanism, PCR genotyping confirmed that the mEGFP single-positive fraction corresponds to EJ-TI alleles (Fig. 1f).

5) Also in Fig. 1e, the EJ-TI donor contains homology arms for both sides of the DSB site at

the genomic target. This could allow integration via single-strand annealing (SSA) or microhomology-mediated end joining (MMEJ), in addition to HDR or c-NHEJ. Consequently, EJ-TI outcomes may include mEGFP-only, mCherry-only, or double-positive cells. The manuscript does not distinguish between these possibilities, which complicates interpretation of the EJ-TI data.

In this study, we clearly distinguish between DNA repair mechanisms (HR, c-NHEJ, SSA, MMEJ, etc.) and the genome-editing outcomes actually observed (knock-in and EJ-TI).

[REDACTED]

Here, knock-in is defined as a precise insertion mediated by homology arms, which corresponds to homology-directed repair (HDR) in the context of genome editing. In contrast, EJ-TI is defined as an outcome in which the entire donor sequence, including both homology arms and the gene of interest, is integrated at the target locus. In our

system, both knock-in and EJ-TI are quantified using the same donor cassette. The mEGFP signal in EJ-TI arises from the donor-encoded splice acceptor, while mCherry is not expressed because EJ-TI formation introduces a stop codon at the 3' end (Fig. 1e). Consequently, EJ-TI events appear as mEGFP-single-positive cells. This configuration was confirmed by PCR analysis, in which only the mEGFP-single-positive fraction yielded the EJ-TI-specific amplicon (Fig. 1f).

The mechanism underlying mEGFP expression in EJ-TI—usage of the donor-derived splice acceptor and termination of mCherry translation by the stop codon—has been incorporated into the revised Results section in response to Comment #4 (page 4, lines 104-114).

- 6) The DSB reporter infers DSB induction from indel frequency. However, it is well-established that initial NHEJ repair of Cas9-induced DSBs is typically accurate, and the observed indels largely result from repeated Cas9 cleavage. Further considered the effect of Cas9 target residency besides Cas9 recleavage on indel formation, the reporter may not reliably measure DSB induction as claimed. The observation that a small fraction (~3%) of BFP-positive cells are RFP-negative also raises questions about this interpretation.

Because no simple, direct, locus-specific, high-throughput assay exists to quantify absolute DSB efficiency in bulk cell populations, we adopted TagBFP loss as a convenient, scalable, and high-probability surrogate that correlates with Cas9-induced editing events. As the reviewer notes, this DSB reporter is a mutation-based proxy rather than a direct physical measure of cleavage; it cannot detect perfectly precise NHEJ or the TAC→TAT substitution at the TagBFP chromophore. In addition, since the DSB reporter (TagBFP) and the TI reporter are integrated at different alleles, DSBs may occur at one reporter locus but not necessarily at the other within the same cell. Therefore, simultaneous cleavage at both sites is not guaranteed, and perfect per-cell correspondence between their readouts cannot be expected.

Despite these limitations, the assay consistently captures on-target DSB induction with high probability: in Fig. 1h, knock-in and EJ-TI are detected only in the TagBFP-negative gate, whereas TagBFP-positive cells show no detectable targeted insertion. Moreover, 97% of TagBFP-positive cells are mCherry-positive. Although a small TagBFP-positive/mCherry-negative fraction (~3%) was observed, the detection probability of the DSB reporter is extremely high, and this limitation does not affect the overall conclusion.

To clarify this limitation, we now explicitly describe it in the Discussion (page 13,

lines 384-392) and, as requested in Comment #11, display DSB efficiency (TagBFP-negative fraction) alongside TI outcomes for the same samples in the revised figures.

- 7) It is well documented that DNA-PKcs inhibition reduces c-NHEJ and enhances HDR. This manuscript reports only a mild decrease in indels and negligible effects on HDR or EJ-TI (Fig. S5b). This unexpected result may be influenced by Cas9 post-cleavage target residency, which affects DSB repair pathway choices (PMID: 35915475). This issue should be further addressed or discussed.

In the DSB reporter, DNA-PK inhibition slightly (but significantly) reduced the fraction of TagBFP-negative cells (Supplementary Fig. 6b). This moderate decrease supports the interpretation that c-NHEJ plays a predominant role in repairing Cas9-induced DSBs, while also implying that alt-EJ pathways can partially compensate when DNA-PK activity is inhibited.

We have revised the Discussion accordingly:

Page 14, Line 398-401: DNA-PK inhibition slightly but significantly reduced the fraction of TagBFP-negative cells, consistent with the dominant contribution of c-NHEJ in DSB repair and suggesting partial compensation by alt-EJ in the DSB reporter.

DNA-PKcs inhibition is widely reported to suppress c-NHEJ and to increase knock-in. In our assays, however, no detectable effect on knock-in or EJ-TI was observed (Supplementary Fig. 6b). Two possible explanations may account for this outcome. First, our experiments were conducted in pluripotent stem cells, which display high baseline HDR/HR activity (PMID: 20446816, 21633706, 29103969); in such HDR-permissive cells, c-NHEJ is relatively less rate-limiting, so its inhibition produces only small shifts in knock-in efficiency. Second, as the reviewer pointed out, pathway usage can differ among target sites depending on Cas9–sgRNA interactions, which may also contribute to the lack of detectable effect observed here. Consistent with these possibilities, previous studies have reported little or no HDR improvement upon inhibition of c-NHEJ factors such as DNA-PKcs (NU7441) or DNA ligase IV (Scr7) in pluripotent stem cells (PMID: 28219395).

Furthermore, to resolve the apparent inconsistency between the ATM- and DNA-PK-inhibitor results—both of which are expected to suppress c-NHEJ, yet only ATM inhibition increased knock-in efficiency—we now clarify this point in the Discussion. In the AAV-donor setting, the primary factor underlying improved knock-in after ATM inhibition is the suppression of ATM-dependent apoptosis in AAV-high cells. Additionally, the effect of c-NHEJ inhibition may become evident only when ATM-

dependent regulation of both DNA-PK activity and 53BP1–RIF1-mediated end protection is simultaneously attenuated.

Accordingly, we have revised the Discussion to clarify these points (page 12, lines 352–358; and page 14, lines 401-408).

- 8) In Fig. 3, linear plasmids should also be included. By directly comparing AAV transduction and circular plasmid electroporation, the authors introduce confounding variables, including differences in delivery method and AAV-specific cellular response, which might contribute to ATM overactivation.

To address the concern about potential confounding effects from delivery methods, we added an “RNP + plasmid-HITI” condition to Fig. 5a (corresponding to Fig. 3a in the original submission). Both AAV transduction and electroporation of the plasmid-HITI markedly increased phosphorylated ATM levels, whereas the circular plasmid donor induced only a slight increase. Together with the data in Fig. 4 and Fig. 5 (showing that empty AAV did not induce ATM activation; Fig. 5b), these results indicate that introduction of linear DNA, rather than AAV-specific effects or differences in delivery method, is responsible for ATM overactivation.

9) In Fig. 4b, TagBFP expression may reflect transfection efficiency more than actual donor copy number. Additional methods are needed to quantify donor DNA copy number more accurately. Moreover, the claim that ATM inhibition increases knock-in by rescuing high-donor-copy cells is based solely on p53 S15 phosphorylation and Caspase-3 activation. Direct measurements of cell viability or apoptosis (e.g., Annexin V/PI staining, live/dead cell assays) would strengthen this conclusion.

To directly verify this proposed mechanism that ATM inhibition enhances knock-in by rescuing AAV-high cells from apoptosis, we newly conducted Annexin V/Zombie NIR flow cytometry and qPCR analysis of intracellular AAV copy number. As a result, ATM inhibition significantly reduced the Annexin V⁺/Zombie NIR⁻ fraction after AAV-mediated genome editing, providing direct evidence that early apoptosis was suppressed by ATM inhibition (Fig. 5f). Furthermore, qPCR quantification of the donor vector, normalized to a single-copy genomic locus, revealed a significant increase in AAV vector genomes per cell upon ATM inhibition (Fig. 6e). These new results directly confirm the proposed sequence—ATM inhibition suppresses AAV-induced apoptosis, thereby increasing per-cell donor copy number and enhancing knock-in efficiency.

10) Earlier studies indicate that ATM is dispensable for HDR in mouse ES cells (e.g., PMID:

23355489, 28659469), yet this manuscript reports a strong HDR enhancement with ATM inhibition. The authors should reconcile these contrasting results in the Discussion.

Our results show that abundant linear donor DNA (including AAV genomes) overactivates ATM signaling. ATM inhibition reduces this response and increases knock-in efficiency. Both cited studies (PMID: 23355489, 28659469) examined chromosomal DR-GFP reporter repair after I-SceI-induced DSBs, where ATM is not overactivated because those experiments did not involve the introduction of abundant linear donor DNA. In those reporter contexts, ATM inhibition decreased HDR. These observations are consistent with our plasmid-donor experiments, in which the circular donor induces only a slight increase in phosphorylated ATM and ATM inhibition decreases knock-in/HDR efficiency. We have added the following sentence to the Discussion:

Page 13, Line 364-366: By contrast, under conditions lacking abundant linear donor DNA (e.g. chromosomal DR-GFP assays), where ATM is not overactivated, ATM inhibition decreases HR^{37, 38}.

11) In the section of Statistical analysis, it is not clear how many independent experiments have been done and whether each independent experiment is performed in triplicates.

Does "n=3" indicated in the manuscript represent 3 independent experiments, each in triplicates, or one single independent experiment in triplicates? In addition, are the efficiencies of NHEJ, HDR-mediated KI, EJ-TI presented in Figures normalized with transfection efficiencies? This information should also be provided along with transfection efficiency values.

We clarified in the figure legends that "n" denotes the number of independent experiments, with the only exception being the AAV-based screening (Supplementary Fig. 7b), which was conducted once.

For efficiency calculations, knock-in and EJ-TI values were normalized to the DSB induction frequency (TagBFP-negative fraction) in each dependent experiment to control for variability in upstream processes such as delivery and cleavage efficiency. This normalization ensures that comparisons accurately reflect differences in downstream repair outcomes. We now display the raw DSB% values together with the normalized knock-in and EJ-TI efficiencies in each relevant figure. The normalization procedure is described in the Methods section (page 19, lines 552-554).

12) To solidify the conclusion that ATM inhibition drives the effects as observed in this manuscript, validation with ATM knockout cells would be highly informative and

strengthen the study.

Given the importance of this point, we performed the requested validation using ATM-knockout cells. Under the same conditions as in Fig. 5b, AAV-induced phosphorylation of p53 and activation of caspase 3 were both attenuated in ATM-knockout cells relative to wild-type cells, showing the same trend as pharmacologic inhibition, although the extent of attenuation was weaker (Fig. 5d; Supplementary Fig. 8b). We interpret this weaker phenotype as a consequence of the loss of ATM's scaffold functions, which likely enables compensatory rerouting of DNA damage signaling. Supporting this interpretation, a previous study showed that kinase-dead ATM, which retains the scaffold but lacks catalytic activity, exhibits a stronger phenotype than ATM-knockout cells (PMID: 23486281). Taken together with this previous report, these findings indicate that the phenotypes observed with ATM inhibitors reflect the specific inhibition of ATM kinase activity, rather than off-target effects of the compounds.

In the revised manuscript, we have incorporated the new immunoblot data from ATM-knockout cells (Supplementary Fig. 8) and updated the text in the Discussion (page 12, lines 337–347).

13) Minor comments: In Fig. 1b. the "LNK" element in the reporter should be explained; In Fig 1c, consider showing the full mutation spectrum for better clarity; It is unclear in the manuscript when the inhibitors were added during genome editing assays.

(a) We now define "LNK" as a linker in the Fig. 1b legend.

(b) We now provide the complete mutation spectrum for the DSB reporter, representing the allele-frequency distribution of InDel sizes at the Cas9 cut site (Supplementary Fig. 1d; corresponding to Fig. 1c). We also show the corresponding data for the TI reporter (Supplementary Fig. 4b). InDel sizes are binned from -30 to +30 bp, with events outside this range aggregated into edge bins. The summary of frameshift and in-frame mutation proportions is retained in Fig. 1c and Supplementary Fig. 4a.

(c) We have clarified the timing of inhibitor treatments in the Methods: inhibitors were added immediately after electroporation and maintained for 24 h until the routine medium change, followed by flow-cytometric analysis at 72 h post-electroporation.

Response to Reviewer #3

We sincerely thank Reviewer #3 for the thorough and constructive evaluation of our manuscript. The thoughtful comments and suggestions were highly valuable and have substantially improved the clarity and depth of our revised version. We have carefully addressed each point as detailed below.

1. The authors convincingly show that inhibition of ATM, and subsequent downregulation of apoptosis, is specific to increasing knock-in efficacy in AAV vectors. However, in figure 4, they use donor vectors to show that increased copy-number increases insertion efficacy, applying this finding to AAV vectors. The authors should perform additional experiments using AAV vectors, to investigate if increased copy number directly decreases targeted insertion via activation of ATM and the apoptotic pathway.

Because reporter expression requires more than 36 h to develop, directly detecting the early selective loss of cells with high donor DNA copy numbers during editing is technically challenging. However, to address this point using AAV vectors, we quantified apoptosis and AAV copy number in the presence of an ATM inhibitor. In AAV-mediated knock-in, ATM inhibition reduced Annexin V⁺/Zombie NIR⁻ apoptotic cells (Fig. 5f) and increased intracellular AAV copy number as measured by qPCR (Fig.

6e). These results obtained with AAV donors indicate that ATM-dependent apoptosis removes cells with high donor DNA copy numbers, and that ATM inhibition prevents such loss, thereby increasing the effective pool of knock-in-competent cells.

This interpretation is further supported by two lines of evidence. First, Fig. 6d shows that ATM activation increases in a dose-dependent manner with increasing AAV input, indicating that cells exposed to high AAV genome loads robustly activate the ATM–p53–caspase pathway. Second, Fig. 6b (plasmid-HITI donor) together with previous AAV-donor work (Rogers 2019; PMID: 31540849) demonstrate that a higher donor load confers greater knock-in competence; in particular, intracellular AAV copy number shows an exceptionally strong correlation with knock-in efficiency ($r^2 > 0.999$ in Rogers).

We have revised the Results and Discussion to incorporate these data obtained using AAV vectors and to clarify the interpretation accordingly (Discussion: page 11, lines 321–332).

2. The authors could explore why RAD51 inhibition promotes an observed decrease in EJ-TI events (as shown in supplementary figure 5) and whether this phenotype is epistatic with the inhibition of PARP1.

We agree with the reviewer's insightful comment. In fact, our long-standing interest lies in understanding how distinct DNA repair pathways give rise to different genome editing outcomes—this question was one of the original motivations for initiating this study.

In this manuscript, single treatments with either a RAD51 inhibitor or a PARP1 inhibitor decreased EJ-TI efficiency, indicating that both RAD51 and PARP1 contribute to this process. This finding was somewhat unexpected and highly intriguing, raising the question of whether it reflects the coexistence of HR- and alt-EJ-related mechanisms or the involvement of repair factors operating in a noncanonical manner.

Comment Redacted

3. One confusing aspect of the manuscript is that the results showing that inhibition of ATM in cells harboring donor vectors enhances survival and knock-in efficacy seems to conflict with earlier findings in the paper showing that activation of ATM improves plasmid-

mediated knock-in efficacy.

We appreciate the reviewer's insightful comment and thank them for highlighting this apparent contradiction, which we had not previously recognized. To clarify this issue, we examined

phosphorylated ATM levels following treatment with a PP2A inhibitor (DMC). As shown in the right panel, PP2A inhibition markedly increased ATM phosphorylation to levels comparable to those observed after AAV transduction (Fig. 5a).

On the other hand, PP2A regulates several distinct DNA damage response (DDR) pathways beyond ATM, and its inhibition can potentially enhance knock-in efficiency by influencing repair pathway selection through multiple mechanisms, some of which act independently of ATM activity.

1. ATR–CHK1 axis: PP2A inhibition sustains CHK1 phosphorylation (PMID: 17015476; 25376608), promoting a cell-cycle state favorable for knock-in (S/G2) and potentially enhancing knock-in efficiency. Consistently, CHK1 phosphorylation following ATR activation (Fig. 2e) improved knock-in efficiency (Fig. 2f,g), supporting the notion that enhanced ATR–CHK1 signaling promotes donor-templated repair.
2. γ H2AX/RPA chromatin signaling: PP2A normally dephosphorylates γ H2AX

and RPA. When PP2A is inhibited, their phosphorylation persists, leading to prolonged DNA damage signaling (γ H2AX: PMID 16310392; RPA: PMID 19704001; reviews: PMIDs 31225502, 31936122). Such sustained signaling can help maintain a checkpoint state permissive for knock-in, similar to the effect of ATR–CHK1 activation.

3. c-NHEJ module (DNA-PKcs/Ku): PP2A directly dephosphorylates inhibitory phosphorylation sites on DNA-PKcs and Ku70/Ku80, thereby maintaining DNA-PK activity. Inhibition of PP2A leads to hyperphosphorylation at these inhibitory sites and reduces DNA-PK protein kinase activity by approximately 50–60% (PMID: 11376007). Therefore, PP2A inhibition dampens the c-NHEJ pathway. Although DNA-PK inhibition alone did not improve knock-in efficiency in our system (Supplementary Fig. 6b), suppression of c-NHEJ by PP2A inhibition may act synergistically with the ATR–CHK1 axis and sustained γ H2AX/RPA signaling described above to promote knock-in.

In summary, PP2A inhibition appears to exert two opposing effects on AAV-mediated genome editing. On one hand, it markedly activates ATM and may further promote apoptosis in cells heavily transduced with AAV, thereby compromising knock-in efficiency. On the other hand, through the mechanisms described above—

including sustained ATR–CHK1 signaling, prolonged γ H2AX/RPA activation, and suppression of c-NHEJ—PP2A inhibition may enhance knock-in efficiency in cells with low to moderate AAV uptake. Because PP2A regulates a wide range of DNA damage response factors, its overall influence on knock-in efficiency is complex and difficult to interpret. Moreover, since the observed increase was not statistically significant ($p = 0.05$), we removed the PP2A data from the revised manuscript to avoid over-interpretation.

4. Better integration of the supplementary material with the main figures and text would make the paper flow better and easier to follow as a reader. For example, the paper would likely be easier to read if some of the supplementary material was integrated into the main figures rather than stand-alone panels in the supplement.

We integrated essential supplementary datasets into the main figures to improve the overall flow of the paper and present each major finding together with its supporting data. Specifically:

- Suppl. Fig. 7 (original submission) → Fig. 2 (revised manuscript; ATR analyses):

This establishes that ATR activity is required for targeted insertion and that activation of the ATR pathway enhances targeted insertion efficiency. In

contrast to ATM, which exerts donor-type-dependent effects, ATR functions as a consistent, donor-type-independent positive regulator. This key distinction is central to the study.

•Suppl. Fig. 8 (original submission) → Fig. 4 (revised manuscript; donor structure):

This compares linear and circular donors to clarify the donor-configuration-dependent ATM phenotype that supports the proposed model.

•Suppl. Fig. 9 (original submission) → Fig. 6d (revised manuscript; donor structure):

This demonstrates AAV dose-dependent ATM phosphorylation, indicating that donor copy number correlates with ATM activation.

Consequently, no stand-alone supplementary panels remain.

5. The authors indicate that they performed a “comprehensive drug screening” but very few details are presented. It is only in the Methods section that they indicate that the compound library used was the DNA damage/repair library. This is critical information to be included in the Results section, otherwise it gives the impression that the authors screened a broader library. In addition, the Methods section should include a separate Compound screening section, where details about the screen should be presented (duration of treatment, concentration, number of compounds, number of sorted cells,

number of replicates of the screen etc). The current paragraph on page 17 is very brief and it is unclear how the screen was done and how robust the assay was.

We revised both the Results and Methods to clarify the scope and procedures of the screen. In the Results (page 6, lines 162–185), we now specify that the DNA Damage/DNA Repair Compound Library (Selleck; 174 compounds) was screened under two donor conditions (plasmid vs AAV) using the triple-reporter ES cells, and that hits were defined as values exceeding the mean of the untreated control ± 3 standard deviations (Average ± 3 SD).

In the Methods (new “DDR-inhibitor screen” subsection; page 20, lines 571–585), we added details of the experimental design, cell plating, compound exposure, flow-cytometric readout, replication, and hit definition. Information on the concentration used for each inhibitor is provided in Supplementary Table 4, as noted in that subsection.

These revisions clarify the exact scope, conditions, and evaluation criteria of the screen and eliminate any impression that a broader, unspecified library was used.

6. The screen results presented in Suppl Fig 6 show a number of additional hits (#49 increases and #144 and #151 decrease the signal in 6a; #54 increases and #33 and #144

decrease the signal in 6b). The authors do not comment on this at all. What are these other hits, and were they validated? The manuscript is narrowly focused on ATM and ATR. Expanding to these other hits could be very powerful.

All additional hits identified in the screen are now explicitly described in the Results section (page 6, lines 162-185).

Briefly, PARP inhibitors showed compound-specific and non-concordant activity: for example, rucaparib phosphate (#49) exhibited an increase in the Cas9-plasmid/plasmid donor condition, whereas other PARP inhibitors—including niraparib (#33) and AZD2461 (#157)—did not reproduce this increase under identical screening conditions. Because the effect failed to reproduce within the PARP-inhibitor class, we did not advance PARP inhibitors for validation in the present study.

In contrast, ATR inhibitors (VE-821 #144 and BAY 1895344 #151) reproducibly decreased knock-in in both plasmid and AAV contexts in the primary screen, whereas ATM inhibitors (KU-55933 #37, KU-60019 #53 ; corrected from #54 in the reviewer's comment) displayed donor-type-dependent behavior. Taken together, although ATR and ATM are two closely related members of the PIKK family that play central roles in DNA-damage signaling, our screen revealed a clear divergence in donor-type responsiveness: ATR acted irrespective of donor type, whereas ATM showed donor-

type dependence. Accordingly, we focused experimental validation on these two kinases (page 7, lines 181-185).

7. Were ATR inhibitors present in the library (one would assume so), and were they identified as hits?

Yes. The library included two ATR inhibitors, and they were identified as negative hits (decreases in knock-in) in both plasmid and AAV donor conditions in the primary screen (VE-821 #144, BAY 1895344 #151). As noted in our response to Comment #6, all additional hits from the screen are now explicitly described in the Results section (page 6, lines 162-185). We also added arrowheads in Supplementary Fig. 7 to indicate the positions of these ATR inhibitors.

8. A significance weakness of the manuscript is that the authors used only small molecule inhibitors to probe the roles of ATM and ATR in these processes. Since these chemicals may have off target effects, it is important for the authors to validate some of their main findings using genetic approaches (gene depletion) to confirm the phenotypes observed using the inhibitors.

In response to the reviewer's important point, we conducted additional validation

using ATM-knockout cells. Under the same conditions as in Fig. 5b, both AAV-induced phosphorylation of p53 and activation of caspase 3 were attenuated in ATM-knockout cells compared with wild-type cells, showing the same trend as pharmacologic inhibition, although the extent of attenuation was weaker (Fig. 5d; Supplementary Fig. 8b). We interpret this weaker phenotype as a consequence of losing ATM's scaffold functions, which may allow compensatory rerouting of DNA damage signaling. Supporting this interpretation, a previous study showed that kinase-dead ATM, which retains the scaffold but lacks catalytic activity, exhibits a stronger phenotype than ATM-knockout cells (PMID: 23486281). Taken together with this previous report, these findings collectively indicate that the phenotypes observed with ATM inhibitors reflect specific inhibition of ATM kinase activity rather than off-target effects of the compounds. We have created a new Supplementary Fig. 8 showing immunoblot data for ATM-knockout cells and expanded the Discussion to clarify this point (page 12, lines 337-347).

For ATR, stable gene depletion is not feasible in mouse ES cells because ATR is essential and ATR-null ES lines cannot be established (PMID: 10801416). We therefore drew on orthogonal pharmacologic and pathway-level data to address specificity (Fig. 2). Two independent ATR inhibitors (VE-821 and ceralasertib) concordantly reduced

knock-in with both plasmid and AAV donors, and activation of the ATR/CHK1 pathway via NRF2 stabilization (KEAP1 inhibition with 4-octyl-itaconate) produced the opposite effect, significantly increasing knock-in efficiency. Taken together, the concordant effects of two ATR inhibitors and the reciprocal increase with pathway activation support on-target ATR involvement. In the revision, we now clearly explain that ATR knockout is not feasible in mouse ES cells due to its essential role, and clarify that specificity was instead supported by two complementary lines of evidence: pharmacologic testing with two ATR inhibitors showing consistent effects, and functional results indicating reciprocal enhancement of knock-in efficiency upon ATR/CHK1 activation induced by KEAP1 inhibition (page 13, lines 378-382).

Response to Reviewer #1

The authors have adequately addressed almost all of my questions with the revision. One minor comment: the statement from Line 364-366, "By contrast, under conditions lacking abundant linear donor DNA (e.g. chromosomal DR-GFP assays), where ATM is not overactivated, ATM inhibition decreases HR^{37,38}", is not fully accurate. In these two cited articles, ATM loss does not affect HDR in mouse ES cells; however, treatment with an ATM inhibitor reduces HDR in the work by Chen et al., who speculate that inactivated ATM kinase by the inhibitor in ATM wild-type mouse ES cells might interfere recruitment of HDR factors to the I-SceI-induced DSBs. This seems to have nothing to do with ATM overactivation.

We thank the reviewer for pointing out this inaccuracy. We removed the original statement and added the Discussion to distinguish pharmacologic inhibition from genetic ATM loss and to reconcile our findings with prior HDR reporter studies, as follows:

Page 13, Line 349-356: This distinction between pharmacologic inhibition of ATM kinase activity and genetic loss of ATM is important when interpreting prior chromosomal HDR reporter studies lacking abundant linear donor DNA (e.g. I-SceI-induced DR-GFP assays), in which genetic loss of ATM had little impact on HDR^{34,35}. By contrast, in our HDR-mediated targeted integration assay using a circular plasmid donor, pharmacologic ATM inhibition markedly reduced knock-in efficiency (Fig. 3a). This apparent discrepancy may reflect fundamental differences between pharmacologic inhibition of ATM kinase activity and genetic loss of ATM, including compensatory rerouting of DNA damage response pathways.